



# Estimation of Evapotranspiration and Other Soil Water Budget Components in an
# Irrigated Agricultural Field of a Desert Oasis, Using Soil Moisture Measurements
Zhongkai Li[a,b,d] , Hu Liu[a,b*] , Wenzhi Zhao[a,b], Qiyue Yang [a,b], Rong Yang [a,b], Jintao Liu[c]
*a Linze Inland River Basin Research Station, Chinese Ecosystem Research Network, Lanzhou 730000, China*
*b. Key Laboratory of Ecohydrology of Inland River Basin, Northwest Institute of Eco-Environment and Resources, Chinese Academy of Sciences, Lanzhou, 730000, China*
*c. State Key Laboratory of Hydrology-Water Resources and Hydraulic Engineering, Hohai University, Nanjing 210098, China*
*d. University of Chinese Academy of Sciences*
*\* The corresponding author, lhayz@lzb.ac.cn*
## Abstract
An accurate assessment of soil water budget components (*SWBCs*) is necessary for improving irrigation strategies in any water-
limited environment. However, quantitative information of *SWBCs* is usually challenging to obtain, because, since the hydrological
process of farmland is principally driven by irrigation (*I*), drainage (*D*), and evapotranspiration (*ET*) in desert oasis settings, none
of the drivers can be easily measured under actual conditions. Soil moisture is a variable that integrates the water balance components
of land surface hydrology, and the evolution of soil moisture is assumed to contain the memory of antecedent hydrologic fluxes,
and thus can be used to determine *SWBCs* from a hydrologic balance. A database of soil moisture measurements from six
experimental plots with different treatments (NT1 to NT6) in the middle Heihe River Basin of China was used to test the potential
of a soil moisture database in estimating the *SWBCs*. We first compared the hydrophysical properties of the soils in these plots, such
as vertical saturated hydraulic conductivity ($K_s$) and soil water retention features, for supporting the *SWBC* estimations. Then we
determined evapotranspiration and other *SWBCs* through a soil-moisture data-based method that combined both the soil water
balance method and the inverse Richards equation. To test the accuracy of our estimation, we used both the indirect methods (such
as power consumption of the pumping irrigation well), plenty of published *SWBCs* values at nearby sites, and the water balance
equation technique to verify the estimated *SWBCs* values, all of which showed a good reliability of our estimation. Finally, the
uncertainties of the proposed methods were analyzed to evaluate the systematic error of the *SWBC* estimation and the restriction for
its application. The results showed significant variances among the film-mulched plots (NT2-6) in both the cumulative irrigation
volumes (between 652.1 mm at NT3 and 867.3 mm at NT6) and deep drainages (between 170.7 mm at NT3 and 364.7 mm at NT6).
Moreover, the unmulched plot (NT1) had remarkably higher values in both cumulative irrigation volumes (1186.5 mm) and deep
drainages (651.8 mm) compared with the mulched plots. Obvious correlation existed between the volume of irrigation and that of
drained water. However, the ET demands for all the plots behaved pretty much the same, with the cumulative ET values ranging
between 489.1 and 561.9 mm for the different treatments in 2016, suggesting that the superfluous irrigation amounts had limited
influence on the accumulated ET throughout the growing season because of the poor water-holding capacity of the sandy soil. This
work confirmed that relatively reasonable estimations of the *SWBCs* in coarse-textured sandy soils can be derived by using soil
moisture measurements; the proposed methods provided a reliable solution during the entire growing season and showed a great
potential for identifying appropriate irrigation amounts and frequencies, and thus a move toward sustainable water resources
management, even under traditional surface irrigation conditions.
## Keywords
Evapotranspiration, Soil water budget, Desert oasis, Soil moisture, Inverse Richards Equation.
## 1. Introduction
Arid inland river basins in Northwestern China are unique ecosystems consisting of ice and snow, frozen soil, alpine vegetation,
oases, deserts, and riparian forest landscapes, in a delicate eco-hydrological balance (Liu et al., 2015). Among these inland basins,
the Heihe river basin (HRB) is one of largest (Chen et al., 2007). The oasis plains in the middle reaches of the HRB have become
an important source of grains, including the largest maize seed production center in China (Yang et al., 2015). Crop water
requirements in this region are supplied mainly by irrigation from the river and from groundwater (Zhou et al., 2017). According to
Wang et al. (2014), agriculture consumes 80 to 90% of the total water resources in the HRB, and has fundamentally altered the
regional hydrological processes and even resulted in eco-environmental deterioration (Zhao and Chang, 2014). Traditional irrigation,
namely flood irrigation in the HRB, has low efficiency (i.e., a high leaching fraction) (Li et al., 2017;Deng et al., 2006) and the



extensive fertilization practices have given rise to higher levels of potential nitrate contamination in the groundwater, because water
and pollutants percolate into the deep sandy soils of the desert oasis, which have low water-holding capacities (Zhao and Chang,
2014). It is crucial to adopt a mechanism that can preserve the role of irrigation in food security, yet with minimal consumption of
the already scarce water, in order to increase water productivity and conservation. Reducing water drainage and thus nitrate
contamination in groundwater, saving water, and increasing water and nitrogen use efficiency, are turning out to be important steps
toward sustainable agriculture in this region (Hu et al., 2008)—steps that are being implemented by developing effective irrigation
schedules (Su et al., 2014).

55       Because allowing the soil to dry out too much may adversely affect the yield and quality of crops, while irrigating too much
can lead to wasted water, loss of fertilizer by leaching, increased operating costs and drainage problems, and sometimes decreased
crop yield or quality (Wright, 1971), an efficient irrigation scheduling program should aim to replenish the water deficit within the
root zone while minimizing leaching below this depth (Bourazanis et al., 2015). Accordingly, an accurate assessment of soil water
budget components ($SWBCs$) is necessary for improving the irrigation management strategies in the oasis fields. However,
quantitative information of $SWBCs$ is usually challenging to obtain (Dejen, 2015). In desert oasis settings, the hydrological process
of farmland is principally driven by irrigation ($I$), drainage ($D$), and evapotranspiration ($ET$). None of these drivers is easily measured
in practice, however. For example, not even the optimal irrigation amount can be determined accurately: the two most common
methods of measuring irrigation water—water meters or indirect methods—pose both economic and operational challenges to water
managers, due to the wide spatial distribution of small fields throughout rural areas (Folhes et al., 2009). Measurement of deep
percolation is also difficult, and reliable data are rare in practice, and thus percolation is often calculated as a residual of the water
balance (Bethune et al., 2008;Odofin et al., 2012). ET is another source of uncertainty inherent in water budget estimations (Dolman
and De Jeu, 2010), and its estimation at the field scale is usually through the application of mathematical models: it is commonly
calculated by relying on reference ET ($ET_0$) or potential ET ($PET$) (Allen et al., 2011;Suleiman and Hoogenboom, 2007;Wang and
Dickinson, 2012;Ibrom et al., 2007).

70       Soil moisture is a variable that integrates the water balance components of land surface hydrology (Rodriguez-Iturbe and
Porporato, 2005), and over time it can be used to develop a record of antecedent hydrologic fluxes (Costa-Cabral et al., 2008). Soil
moisture measurements were used to estimate the infiltration for unsaturated porous mediums by numerical solutions as early as the
1950s (Hanks and Bowers, 1962;Gardner and Mayhugh, 1958). With the advent of automated soil moisture monitors (Topp et al.,
1980), ET estimation was implemented using continuous soil moisture data by simple water balance approaches (Young et al., 1997),
but the computations are usually interrupted during rainfall or irrigation periods, as there is no means of accounting for drainage or
recharge, due to inadequate turbulent flux measurements (Naranjo et al., 2011). It has only been during recent years that some
researchers, including Schelde et al. (2011) and Guderle and Hildebrandt (2015), have started exploring the potential of using highly
resolved soil moisture measurements to determine ET and sink term profiles, by accounting for vertical flow, demonstrating that
such measurements can work when the appropriate approach is used. Rahgozar et al. (2012) and Shah et al. (2012) extended these
methodologies to determine other components of the water budget, such as lateral flow, infiltration, interception capture, storage,
surface runoff, and other fluxes. During the last 30 years, Time Domain Reflectometry (TDR) has become quite common and popular
for measuring volumetric soil moisture content around the world (Kirnak and Akpinar, 2016). For example, it is being used more
and more frequently for monitoring soil moisture dynamics of agro-ecosystems in both the Chinese Ecosystem Research Network
(CERN) and the U.S. Long-Term Ecological Research Network (US-LTER) (Fu et al., 2010;Sr et al., 2003), because of its flexibility
and accuracy (Schelde et al., 2011). Also, with this processes, methods based on soil moisture data have become one of the most
promising ways to quantify $SWBC$ information in different ecosystems (Li et al., 2010). So far, however, almost no works have been
published on testing the potential of using a soil moisture database as a method to systematically estimate all the $SWBCs$ of farmland
in dry lands, including the desert oasis of the middle HRB (Liu et al., 2015), where the principal soils are coarse-textured (Grayson
et al., 1999;Yang et al., 2018b). As one of the efforts in this region, intensive TDR measurements of soil moisture were conducted
in a long-term field experiment that was originally designed to test the accumulative impacts of different cropping systems (i.e.,
maize and alfalfa) and agronomic manipulation (i.e., succession cropping, crop rotation, row intercropping) on soil property
evolution in the ecotones of desert and oasis. Within the context of the largest-scale deployment of soil moisture monitoring system
in the world, exploring a reliable farmland $SWBC$ estimation model, which can make the most of the vast amounts of soil moisture





data, is crucial for irrigation management optimization (Musters and Bouten, 2000;Sharma et al., 2017), especially for irrigating
arid regions with coarse-textured soils.
Based upon a soil moisture database, as mentioned above, this work aimed 1) to investigate the performance of using soil
moisture measurements to determine *ET* and other *SWBCs* in the croplands of a desert oasis, serving as a framework for farmland
*SWBC* estimation for coarse-textured soils; 2) to estimate the effects of different cropping systems and agronomic histories, on the
hydrophysical soil properties, and to discuss these effects on the practical application of our method in different fields; and 3) to
determine the potential for using a soil-moisture data-based method to improve irrigation strategies in a desert oasis.

## 2.    Materials and Methods

### 2.1 Study area

The study sites were located in the transition zone between the Badain Jaran Desert and the Zhangye Oasis in the middle HRB
(Fig. 1). More specifically, they were in the Linze Inland River Basin Research Station of the Chinese Academy of Science (39º21'N,
100º17'E, altitude 1382m). This region has a temperate continental desert climate. The annual average temperature is about 7.6ºC,
and the lowest and highest temperatures are -27ºC and 39.1ºC for winter and summer, respectively. The annual average precipitation
is 117 mm and the mean potential evaporation is about 2,366 mm/a. The annual dryness index is 15.9. About 60% of the total
precipitation, with low rainfall intensity, is received during July–September, with only 3% occurring during winter. Northwest winds
prevail throughout the year, with intense sandstorm activity in spring. This region was part of a sandstorm-eroded area, and the
research site was converted into an artificial oasis during the 1970s. As a result, the soil types are dominated by sandy loam and
sandy soil, and characterized by coarse texture and rapid infiltration (Zhao et al., 2010). The local dominant species are *Scotch Pine,*
*Gansu poplar, wheat,* and *maize* (Liu et al., 2015), and sand-fixation plant species (planted since the 1970s), include *Haloxylon*
*ammodendron*, *Elaeagnus angustifolia*, *Tamarix ramosissima*, *Nitraria sphaerocarpa*, and annual herbaceous species such as *Bassia*
*dasyphylla*, *Halogeton arachnoideus*, *Suaeda glauca* and *Agriophyllum squarrosum*. The growing season of these plants and forages
usually starts in early April and normally continues through the month of September (DOY 94-288 Julian days >0ºC).




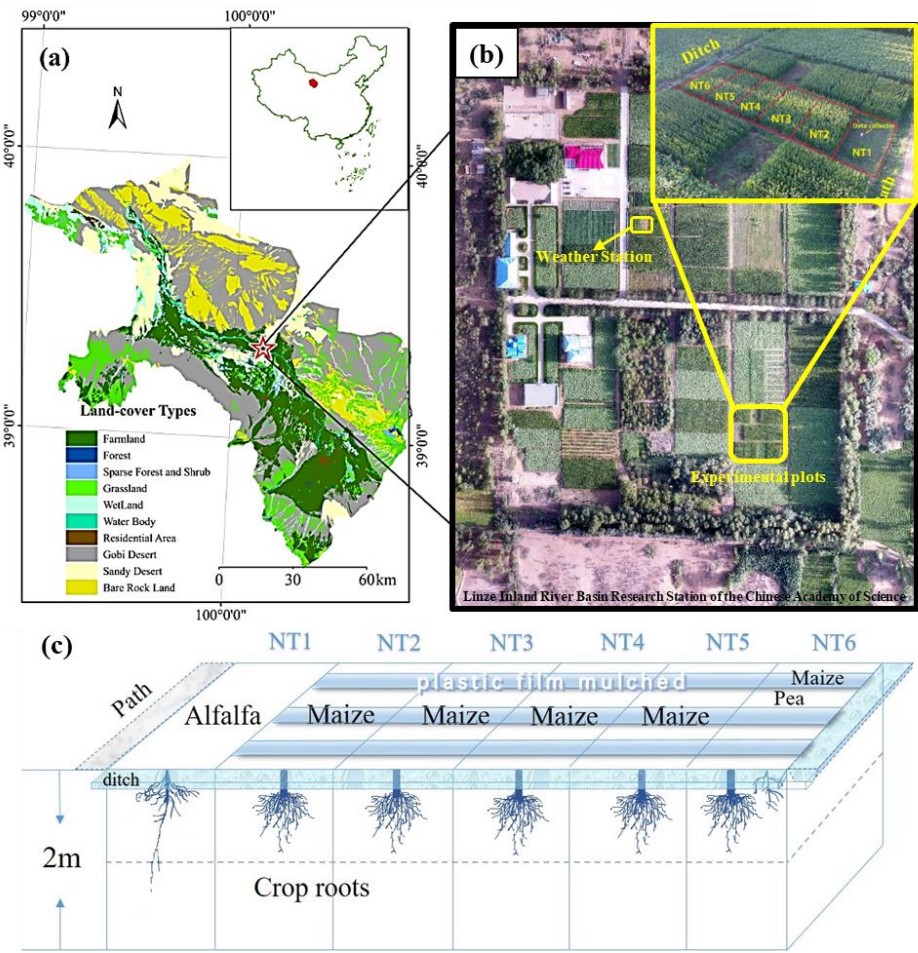


**Figure 1. a)** *Map of study area and research site;* **b)** *aerial view of the study site;* **c)** *detailed design of the field experiments in 2016*
## 2.2 Site description
In order to investigate the accumulative effect of different cropping systems and agronomic manipulation on soil property
evolution, a long-term field experiment with six different treatments was set up in 2007. The experiment was performed with
randomized complete block design (RCBD) with three replications (Figs. 1b and 1c), so that in total, 18 plots of 6m × 9m were
established. We assumed that the soil texture and cultivation history (about 40 years) of the plots subjected to the different treatments
were essentially identical before the experiment was conducted. The middle area of the three replications (6 plots, NT1 to NT6) was
selected for installing the TDR sensors. The applied treatments of NT1 to NT6 were sequentially as follows: (1) continuous pasture
cropping; (2) continuous maize cropping; (3) continuous maize cropping with straw return; (4) maize-maize-pasture rotation; (5)
maize-pasture rotation; (6) maize-pasture intercropping. Plastic film mulching was applied during the initial growing season, and
the irrigation method was furrow irrigation (Zhao et al., 2015). In 2016, NT1 was planted in alfalfa without plastic film mulch; NT2
to NT5 in maize with plastic film mulch; and NT6 in interlaced maize (mulched) and peas (non-mulched) (Fig. 1c). Maize and peas
are annual crops, and about 80% of the maize roots were distributed in the soil layers between 0 and 40 cm. only a few maize roots
can reach 100 cm, while pea roots are usually found within 30-cm depth. Alfalfa is a perennial forage legume which normally lives
four to eight years, and about 70% of alfalfa roots were distributed in the soil layers between 0 and 30 cm; only a few alfalfa roots
can reach 110 cm in the sandy soils of this region (Sun et al., 2008). The growing season of maize and alfalfa in the region is usually
from early April until late September (Zhao and Zhao, 2014). Alfalfa was harvested twice during the growing season of 2016.





Harvest 1 was conducted on 16 July, and the subsequent re-growth was harvested on 28 September (Su et al., 2010).
The groundwater table depth fluctuated from 5 to 8 m at the experimental field during the year 2016. Irrigation with water
extracted from a nearby pumping irrigation well was applied one by one in the plots from NT6 to NT1 during each irrigation event,
and this work was usually completed in 3 hours or less. The power consumption of the pumping irrigation well was recorded as an
in-situ observation to obtain the actual total irrigation amount of all plots through a well-built relationship at field scale: i.e., it
obtained the average actual irrigation amount of the six plots. The volumetric soil moisture of the six plots (NT1 to NT6) was
measured with a TDR system (5TE, Decagon Devices Inc. Pullman, WA, USA), which were installed at 5 different depths (20, 40,
60, 80, and 100 cm) at each plot, with measurement intervals of 10 minutes. Before use, the TDR was calibrated from soil columns
in the laboratory with known volumetric water content ($\theta_v$). A maximum likelihood fitting procedure was used to correct the
observed data to eliminate the potential errors induced by the soil texture and salinity (Muñoz-Carpena, 2004). Soil bulk density
($\rho_b$), vertical saturated hydraulic conductivity ($K_s$), and soil water retention were determined using standard laboratory procedures
on undisturbed soil cores in steel cylinders (110 cm$^3$ in volume, 5 cm in height) taken at 20-cm intervals down to 100-cm depth.
Soil water retention curves were measured at the pressure heads of -0.01, -0.05, -0.1, -0.2, -0.4, -0.6, -0.8, -1, -2, -5, -10, -15, -20,
and -25 bars. $K_s$ was measured with an undisturbed soil core using the constant head method, i.e., measured 36 h after saturated
water flow at a constant head gradient (5 cm) (Salazar et al., 2008). The values of field capacity ($\theta_{fc}$) and wilting point ($\theta_w$) were
empirically related to the corresponding soil water (matrix) potentials through the determined soil-water retention curves (-0.1 bar
for $\theta_{fc}$ and -15 bar for $\theta_w$). Hourly climatic data, including precipitation, temperature, radiation, wind, and potential evaporation
were recorded by a weather station located about 150 meters away from the experimental site (Fig. 1).
## 2.3 Calculation methods
**1) Water storage and irrigation amounts**
Soil water storage ($S$) was calculated for the soil depth within the root zone (0-110 cm) based on the sensor readings through
the equation:
$$S = \sum_{i=1}^{5} \theta_i Z_i' \qquad (1)$$
where $\theta_i$ is the soil moisture of layer $i$; and $Z_i'$ is the layer thickness between 10cm above and 10cm below the sensor installation
depth (except for the top 30-cm soil layer, which is represented by the TDR installed at 20cm). At the field level, examples of
inflows are irrigation and rainfall, and examples of outflows are evaporation and deep leakage beyond the root zone. An irrigation
event usually lasted 20 to 30 minutes in each of the independent plots depending on the growth stages of the plants. Soil moisture
increased rapidly following irrigation events and decreased quickly as well during the subsequent dry-down period. Rapid drying
usually occurs for a few hours after a soil has been thoroughly wetted because of high water conductivity (Fig. 2). The preferential
flow was neglected in the selected soil profiles because the larger hydraulic conductivity of sandy soil itself neutralizes the effects
of preferential flow, and because coarse soil is relatively inimical to the formation of stable preferential flow paths (Hamblin, 1985).
Because the relatively short irrigation times that hampered the form of the steady infiltration rate (Bautista and Wallender, 1993;Selle
et al., 2011), we hypothesized that no surface-water excess or steady-state flow took place during any irrigation event, and assumed
that deep percolation began after soil moisture storage reached maximum ($S_{max}$); thus the irrigation volume ($V$) could be calculated
as the difference between $S_{max}$ and $S_{ini}$:
$$V = S_{max} - S_{ini} \qquad (2)$$
where $S_{max}$ is the maximum soil water storage of the root zone (0-110cm) after one irrigation event began and $S_{ini}$ is the initial
soil water storage of the root zone before irrigation (Figure 2). Although the deep percolation of NT2 in this irrigation event had
begun before its soil moisture storage reached maximum (Fig. 2b), it had little effect on the estimation of irrigation volume because
the maximum soil water storage differed little (by only 1.86 mm) before and after deep percolation began. We checked all sixty of
the irrigation events of NT1-NT6 during the entire growing season period, and there were no underestimates of $S_{max}$ except for
two irrigation events in NT2, which had a slight underestimates of 1.86 mm and 10.3 mm, which generated errors of 1.1% and 4.1%,
respectively.





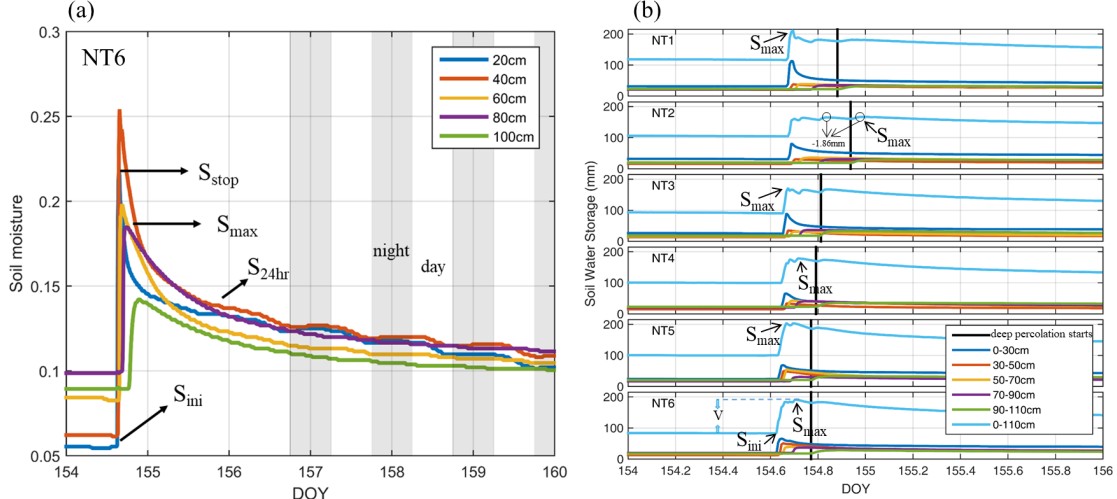

**Figure 2. (a)** *Example diagram of the volumetric soil water content at various depths of NT6 during and after the irrigation event of 107.1 mm on DOY 154-160 (2016). $S_{stop}$: irrigation event ends, and moisture of uppermost soil layer starts to decrease; $S_{max}$: maximum water storage; $S_{24hr}$: deep percolation ends one day later; after this point, ET dominates the water-loss processes; $S_{ini}$: pre-irrigation, soil moisture minimum. The gray stripes between 156-160 DOY represent nights, i.e., 6:00 pm to 6:00 am of the next day.* **(b)** *Verification of the assumption of equation 2, i.e., that $S_{max}$ appeared before deep percolation began, during the irrigation event on DOY 154-156 (2016). The black solid line represents the time that deep percolation began in each plot (NT1-6).*

**2) Drainage and evapotranspiration**

Following irrigation water applications, the drainage behavior of the soils consisted of two stages: 1) rapid drainage and 2) slow drainage. During irrigation, the root zone became effectively saturated, and rapid drainage followed, leading to deep percolation. Then, as the water content in the soil fell, the hydraulic conductivity decreased sharply, as did the rate of drainage. The second phase, slow drainage, may continue for several days or months, depending on the soil texture (Bethune et al., 2008). We assumed that rapid drying or drainage ceased 24 hours after an irrigation event, and thus rapid drainage ($Q_1$) could be estimated through the variances of water storage and actual ET during the period (Eq. 3). The actual ET during the period was assumed to be equal to the potential ET, because ET occurs unhindered under non-water-stress conditions.

$$Q1 = S_{max} - S_{24hr} - ET_p \qquad (3)$$

where $S_{24hr}$ is the soil moisture storage 24 hours after irrigation; $S_{max}$ is the maximum water storage after irrigation; and $ET_p$ is the potential ET calculated with the Penman-Monteith combination equation during that day.

Slow drainage is especially important for sandy soils (Bethune et al., 2008), as along with ET, it dominants the water loss processes during the second drying stage before the next irrigation event. Following Zuo et al. (2002) and Guderle and Hildebrandt (2015), an inverse method was employed to estimate the slow drainages and the average root water uptakes by solving the mixed theta-head formulation of the 1-D Richards Equation (Eq. 4) and iteratively searching for the sink term profile that produces the best fit between the numerical solution and the measured values of soil moisture content. ET is then obtained by summing rainfall and the sink term ($S_p$), and the drainage for this period is estimated as the water flux across the lower boundary of the soil profile. The above-mentioned 1-D Richards Equation is written as:

$$C(h)\frac{\partial h}{\partial t} = \frac{\partial}{\partial t}\left[K(h)\left(\frac{\partial h}{\partial z} - 1\right)\right] - Sp(z,t); \qquad (4)$$

$$h(z,0) = h_0(z) \qquad 0 \le z \le L; \qquad (5)$$

$$\left[-K(h)\left(\frac{\partial h}{\partial z} - 1\right)\right]_{z=0} = -E(t) \qquad t > 0; \qquad (6)$$

$$h(L,t) = h_l(t) \qquad t > 0; \qquad (7)$$

where h is the soil matric potential (cm); C(h) the soil water capacity (cm$^{-1}$); K(h) the soil hydraulic conductivity (cm d$^{-1}$); $h_0(z)$ the initial soil matric potential in the profile (cm); E(t) the soil surface evaporation rate (cm); $h_l(t)$ the matric potential at the lower boundary (cm); L the simulation depth (cm); and z the vertical coordinate originating from the soil surface and moving positively





downward (cm). The iterative procedure runs the numerical model over a given time step ($\Delta t$) in order to estimate the soil water
content profile $\tilde{\theta}_i^{v=0}$ at the end of the time step, assuming that the sink term $\widetilde{Sp}_{im,i}^{(v=0)}$ is zero over the entire profile at the beginning,
where ~ depicts the estimated values at the respective soil layer $i$, and $v$ indicates the iteration step. Next, the sink term profile
$\widetilde{Sp}_{im,i}^{(v=1)}$ is set equal to the difference between the previous approximation $\tilde{\theta}_i^{v=0}$ and the measurements $\theta_i$, while accounting for
soil layer thickness and the length of the time step for units. In the following iterations, $\tilde{S}p_{im,i}^{(v)}$ was used with the Richards equation
to calculate the new soil water content $\tilde{\theta}_i^v$. The new average sink term $\widetilde{Sp}_{im,i}^{(v+1)}$ was then determined with Eq. (8):
$$\widetilde{Sp}_{im,i}^{(v+1)} = \widetilde{Sp}_{im,i}^{(v)} + \frac{\tilde{\theta}_i^v - \theta_i}{\Delta t} \cdot d_{z,i}; \qquad (8)$$

A backward Euler with a modified Picard iteration finite differencing solution scheme was adopted to inversely obtain the
solution, and this implementation follows exactly the algorithm outlined by Celia et al. (1990). Three steps proposed by Guderle
and Hildebrandt (2015) were taken to determine when the iteration process could be terminated in this calculation:
a.  Evaluate the difference between the estimated and measured soil water contents (Eq. 9) and compare the change in this
difference to the difference from the previous iteration (Eq. 10):
$$e_i^{(v)} = |\theta_i - \tilde{\theta}_i^v| \qquad (9)$$
$$\varepsilon_{GH,i}^{(v)} = |e_i^{(v-1)} - e_i^{(v)}| \qquad (10)$$

b.  In soil layers where $\varepsilon_{GH}^{(v)} < 0$, set the root water uptake rate back to the value of the previous iteration $\widetilde{Sp}_{im,i}^{(v+1)} = \widetilde{Sp}_{im,i}^{(v-1)}$, since
the current iteration was no improvement. Only if $\varepsilon_{GH}^{(v)} \geq 0$, go to the next step.
c.  If $e_i^{(v)} > 1 \times 10^4$, calculate $\widetilde{Sp}_{im,i}^{(v+1)}$ according Eq. (8); otherwise the current iteration sink term ($\widetilde{Sp}_{im,i}^{(v+1)} = \widetilde{Sp}_{im,i}^{(v)}$) is retained,
as it results in a good fit between estimated and measured soil water content.

### 3) Boundary setting and data collection

To reduce computational complexity, uniform soil profiles were assumed because there were no significant stratification
differences within the sandy soils (Table 2) (Liu et al., 2015). The upper boundary of the calculation was set as the atmospheric
boundary condition, and the calculation involved actual precipitation, irrigation, and potential evapotranspiration rates calculated
through Penman-Monteith combination equations using hourly environmental data during the growing season of 2016 (Fig. 3). The
meteorological measurements were monitored at the nearby weather station (150 m away from our study plots, Fig. 1), which had
the same underlying surface as the experimental plots (Fig. 1b), and were used to compute the upper boundary condition. The film
mulching effects on the upper boundary condition were modeled as proportionally damped $ET_{p,a} = \beta \times ET_p$, where $\beta$ is the area
percentage without plastic film mulching in each experimental plot (i.e., 60%), and $ET_p$ is the potential ET. For coding convenience,
the bare soil evaporation ($E_a$) was determined through a simplified method proposed by Porporato et al. (2002): i.e., the evaporation
was assumed to linearly increase with soil moisture ($\theta$) from 0 at the hygroscopic point ($\theta_h$), to $E_{p,a}$ at the field capacity ($\theta_{fc}$). For
values $\theta$ exceeding the field capacity, evapotranspiration was decoupled from soil moisture and remained constant at $E_{p,a}$.
However, we did not set specific upper boundaries for inter-cropping treatments, because the difference in surface soil evaporation
between mono- and inter-cropping treatments was relatively small when compared with the transpiration over a growing season.
The surface fluxes were incorporated by using the average hourly rates, distributed uniformly over each hour. The lower boundary
condition was set as a soil matric potential boundary because the groundwater table depth (deeper than 3.5 m) was far below the
crop effective root depth during the growing season, and any capillary rise from groundwater could be ignored in this study. A unit
vertical hydraulic gradient boundary condition (i.e., $h = -5cm$) was implemented in the simulation in the form of a variable flux
boundary condition. The drainage rate $q(n)$ assigned to the bottom node $n$ was determined by the software as $q(n) = -K(h)$, where
$h$ is the local value of the pressure head and $K(h)$ is the hydraulic conductivity corresponding to this pressure head (Odofin et al.,
247 2012).

We used soil moisture dynamics measured in the soil profiles as inputs to inversely solve for sink term profiles at each plot for
each hour (Lv, 2014). The soil moisture measurements for 10-minute intervals during the period were hourly averaged to numerically
filter out the noise associated with highly resolved data. This had the effect of slightly reducing the infiltration and ET estimates,
but this effect in the overall results is negligible, according to Guderle and Hildebrandt (2015). The actual amount of water delivered
for irrigation ($Q_0$) was determined from the power consumption of water pumping ($P_0$), through a relationship established between
the two: $Q_0 = P_0 \times \eta$, where $\eta$ is the ratio of the power consumption per unit water pumped and is likely to be different for
different pumping heads. The coefficient was experimentally determined to be 8.5 $m^3 kW^{-1} h^{-1}$ for a head corresponding to 0.95



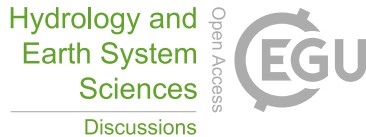

kg/cm$^2$ of delivery pressure, in this study.

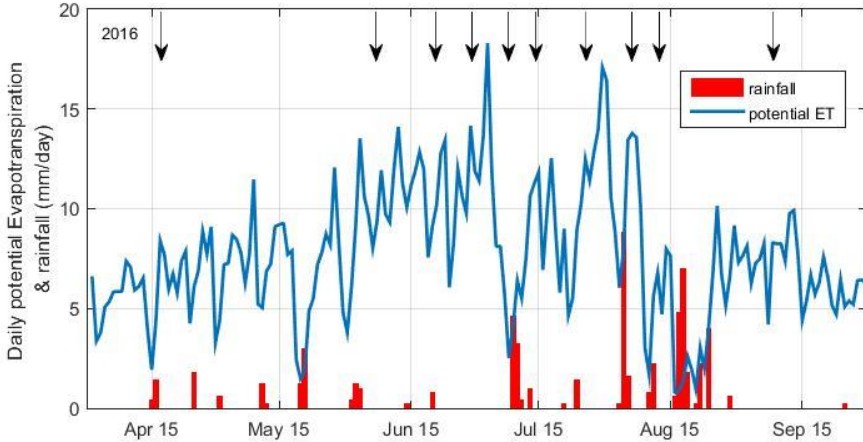


**Figure 3.** *Measured daily rainfall and potential ET estimated with the Penman-Monteith method during the growing season of 2016 at Linze*
*Station. The cumulative rainfall during the growing season was 69.2mm in 2016, and the black down arrows represent irrigation events.*

***Table 1.*** *Nomenclatures involved in this study*

| | | | |
|---|---|---|---|
| $V$ | irrigation amount for one irrigation event (mm) | $K(h)$ | soil hydraulic conductivity (cm d$^{-1}$) |
| $S$ | soil water storage (mm) | $h_0(z)$ | initial soil matric potential in the profile (cm) |
| $S_{stop}$ | soil moisture storage when irrigation was stopped (mm) | $E(t)$ | soil surface evaporation rate (cm) |
| $S_{ini}$ | soil moisture storage before irrigation start (mm) | $h_l(t)$ | matric potential at the lower boundary (cm) |
| $S_{24hr}$ | soil moisture storage 24 hours after irrigation (mm) | $L$ | simulation depth (cm) |
| $S_{max}$ | maximum soil water storage during irrigation event (mm) | $z$ | vertical coordinate originating from the soil surface and moving positively downward (cm) |
| $\theta_i$ | volumetric soil water content of layer $i$ | $\tilde{\theta}_i^{v=0}$ | soil water content profile of soil layer $i$ at the beginning of each calculation |
| $\theta_v$ | theoretical volumetric water content calculated by the ratio of soil volume to water volume | $\widetilde{Sp}_{im,i}^{(v=0)}$ | sink term of soil layer $i$ at the beginning of irrigation, assuming it is zero |
| $\eta$ | ratio of the power consumption per unit water pumped | $d_{z,i}$ | thickness of soil layer $i$ |
| $t$ | time | $\sim$ | estimated values at soil layer $i$ |
| $Q$ | steady-state drainage (mm) | $v$ | iteration step |
| $ET_p$ | potential ET during irrigation day (mm) | $\tilde{\theta}_i^v$ | soil water content of step $v$ |
| $Z'_i$ | detection range of TDR, i.e., 20 cm | $\widetilde{Sp}_{im,i}^{(v)}$ | average sink term of step $v$ |
| $Sp$ | sink term, i.e., water extraction by roots, evaporation, etc. (cm) | $\Delta t$ | given time step |
| $h$ | soil matric potential (cm) | $\varepsilon_{GH,i}^{(v)}$ | difference between $e_i^{(v-1)}$ and $e_i^{(v)}$ |
| $C(h)$ | soil water capacity (cm$^{-1}$) | $e_i^{(v)}$ | difference between estimated and measured soil water content |
| $Q_0$ | real amount of water delivered for irrigation (m3) | $P_0$ | power consumption (kWh) |
| $D_{seas}$ | theoretical drainage volume over entire growing season in 2016 (mm) | $R_{seas}$ | cumulative rainfall during entire growing season in 2016 (mm) |
| $V_{seas}$ | theoretical irrigation volume over entire growing season in 2016 (mm) | $ET_{seas}$ | theoretical ET volume during entire growing season in 2016 (mm) |
| $\Delta S$ | difference in soil water storage before and after the growing season (mm) | $\rho_b$ | soil bulk density (g/cm$^3$) |
| $K_s$ | saturated water conductivity (cm/day) | $\theta_s$ | saturated water content |
| $\theta_{fc}$ | field capacity | $\theta^*$ | water stress point |
| $\theta_w$ | wilting point | $\Psi$ | soil water (matric) potential |
| $\theta_h$ | hygroscopic point | $\beta$ | the area percentage without plastic film mulching |
| $E_a$ | bare soil evaporation | $E_{p,a}$ | bare soil evaporation when soil moisture at field capacity |


## 3. Results
### 3.1 Soil hydrophysical properties
An accurate measurement of soil hydraulic parameters is crucial for this inverse method and is helpful in explaining the
movement of soil water flow. A summary of the most important soil hydrophysical characteristics of the soils at 0–100-cm depth
(NT1 to NT6, and two other representative fields) in relation to their capacity for water storage is listed in Table 2. The textures





were largely loamy sandy in the plots NT1-NT6, in contrast to the sandy loam soil in an old oasis field with a long tillage history
(~100 years) and sandy soil in the desert with no tillage history (Table 2). Their bulk densities were generally between 1.4 and 1.5
g/cm³—slightly higher than that in the local desert land, but still lower than that in maize fields of the old oasis. $\theta_s$, $\theta_{fc}$ and $\theta_w$
of the plots showed the same tendency of increasing soil hydrophysical properties (toward better water retention) as the bulk
densities (Table 2). However, those parameters of the soil profiles are very similar to each other, especially between the same soil
depths (horizontal) of the plots, suggesting that the different planting systems had similar influences on the soil hydrophysical
proprieties, at least at the scale of 10 years. The effects of different cropping systems on soil moisture release characteristics are
shown in Fig. 4. As expected, the relationship between soil water potential and volumetric water content across all data and treatment
combinations followed a curvilinear pattern, where the water potential increased exponentially as soil water content increased.
The large and varying values of saturated drainage velocity ($K_s$) showed a great drainage potential in the coarse-textured soil
and an obvious heterogeneity in both horizontal and vertical profiles across the six plots (Table 2). Soil moisture characteristic
curves (SMC) in the six profiles are shown in Fig. 4, which indicates almost the same soil water content for all the plots, NT1-NT6,
under the same suction head; i.e., all the soil profiles were nearly saturated when the water potential reached the -0.01 bar and little
was available after the soil water potential dropped to the -15 bar. Two obvious inflection points were observed, at $\theta \cong 0.08$ and
0.3, $\psi \cong -0.32$ and -15.2 bar in each of the soil moisture characteristic curves from NT1-NT6.
The slopes of the soil water potential-moisture, especially the parts between the inflection points of the six plots, were very close to
each other, and also similar to that of the desert soil, suggesting similarly poor water capacities of the sandy soils (Sławiński et al.,
2002). A very significant difference in water capacities was observed when comparing the SMC of NT1-NT6 with that of the old
oasis field, indicating that a considerably long period of time is still needed, for high soil water capacity to evolve, for these
experimental sites.

*Table 2. Soil physical characteristics in the six experimental plots and two other selected plots around the study site*

| | NT1 | | | | | NT2 | | | | | NT3 | | | | | NT4 | | | | |
|---|---|---|---|---|---|---|---|---|---|---|---|---|---|---|---|---|---|---|---|---|
| | $K_s$ | $\rho_b$ | $\theta_s$ | $\theta_{fc}$ | $\theta_w$ | $K_s$ | $\rho_b$ | $\theta_s$ | $\theta_{fc}$ | $\theta_w$ | $K_s$ | $\rho_b$ | $\theta_s$ | $\theta_{fc}$ | $\theta_w$ | $K_s$ | $\rho_b$ | $\theta_s$ | $\theta_{fc}$ | $\theta_w$ |
| 20 cm | 47.2 | 1.38 | 0.36 | 0.25 | 0.09 | 183 | 1.46 | 0.34 | 0.19 | 0.08 | 44.3 | 1.40 | 0.36 | 0.21 | 0.09 | 54.1 | 1.39 | 0.38 | 0.21 | 0.08 |
| 40 cm | 46.8 | 1.55 | 0.33 | 0.21 | 0.06 | 82.1 | 1.55 | 0.32 | 0.15 | 0.05 | 259 | 1.54 | 0.34 | 0.18 | 0.06 | 266 | 1.50 | 0.36 | 0.17 | 0.06 |
| 60 cm | 166 | 1.48 | 0.35 | 0.20 | 0.06 | 118 | 1.53 | 0.34 | 0.20 | 0.05 | 73.8 | 1.53 | 0.35 | 0.19 | 0.05 | 355 | 1.47 | 0.36 | 0.16 | 0.06 |
| 80 cm | 61.0 | 1.45 | 0.33 | 0.17 | 0.05 | 164 | 1.48 | 0.35 | 0.18 | 0.05 | 1007 | 1.46 | 0.35 | 0.18 | 0.05 | 192 | 1.47 | 0.35 | 0.20 | 0.06 |
| 100 cm | 273 | 1.46 | 0.34 | 0.18 | 0.05 | 99.7 | 1.49 | 0.34 | 0.15 | 0.05 | 46.1 | 1.44 | 0.35 | 0.16 | 0.05 | 80.0 | 1.40 | 0.37 | 0.23 | 0.06 |
| | | | | | | | | | | | | | | | | | | | | |
| $\bar{X}$ | 119 | 1.46 | 0.34 | 0.20 | 0.06 | 129 | 1.50 | 0.34 | 0.17 | 0.06 | 286 | 1.47 | 0.35 | 0.18 | 0.06 | 189 | 1.45 | 0.36 | 0.19 | 0.06 |
| $SD$ | 99.6 | 0.06 | 0.01 | 0.03 | 0.02 | 42.8 | 0.04 | 0.01 | 0.02 | 0.01 | 413 | 0.06 | 0.01 | 0.02 | 0.01 | 126 | 0.05 | 0.01 | 0.03 | 0.01 |
| | NT5 | | | | | NT6 | | | | | Maize field in old oasis | | | | | Local desert land | | | | |
| | $K_s$ | $\rho_b$ | $\theta_s$ | $\theta_{fc}$ | $\theta_w$ | $K_s$ | $\rho_b$ | $\theta_s$ | $\theta_{fc}$ | $\theta_w$ | $K_s$ | $\rho_b$ | $\theta_s$ | $\theta_{fc}$ | $\theta_w$ | $K_s$ | $\rho_b$ | $\theta_s$ | $\theta_{fc}$ | $\theta_w$ |
| 20 cm | 121 | 1.42 | 0.37 | 0.24 | 0.09 | 89.6 | 1.50 | 0.32 | 0.25 | 0.09 | 28.8 | 1.61 | 0.38 | 0.29 | 0.11 | 42.5 | 1.46 | 0.36 | 0.16 | 0.05 |
| 40 cm | 168 | 1.46 | 0.34 | 0.19 | 0.07 | 575 | 1.53 | 0.33 | 0.20 | 0.06 | 20.2 | 1.61 | 0.37 | 0.28 | 0.12 | 48.1 | 1.46 | 0.35 | 0.17 | 0.05 |
| 60 cm | 41.3 | 1.39 | 0.40 | 0.29 | 0.09 | 66.5 | 1.45 | 0.37 | 0.18 | 0.05 | 37.4 | 1.56 | 0.38 | 0.28 | 0.10 | 30.9 | 1.44 | 0.39 | 0.20 | 0.07 |
| 80 cm | 38.3 | 1.49 | 0.37 | 0.21 | 0.05 | 331 | 1.50 | 0.34 | 0.18 | 0.04 | 76.3 | 1.59 | 0.37 | 0.24 | 0.09 | 33.3 | 1.45 | 0.33 | 0.18 | 0.05 |
| 100 cm | 671 | 1.47 | 0.34 | 0.19 | 0.06 | 18.6 | 1.47 | 0.35 | 0.14 | 0.04 | 47.5 | 1.58 | 0.40 | 0.29 | 0.12 | 26.9 | 1.43 | 0.28 | 0.17 | 0.03 |
| | | | | | | | | | | | | | | | | | | | | |
| $\bar{X}$ | 208 | 1.45 | 0.36 | 0.22 | 0.07 | 216 | 1.49 | 0.34 | 0.19 | 0.06 | 42 | 1.59 | 0.38 | 0.28 | 0.11 | 36 | 1.45 | 0.34 | 0.17 | 0.05 |
| $SD$ | 265 | 0.04 | 0.02 | 0.04 | 0.02 | 234 | 0.03 | 0.02 | 0.04 | 0.02 | 22 | 0.02 | 0.01 | 0.02 | 0.01 | 9 | 0.01 | 0.04 | 0.02 | 0.01 |

$K_s$: saturated water conductivity (cm/day); $\rho_b$: bulk density (g/cm³); $\theta_s$: saturated water content (100%); $\theta_{fc}$: field capacity (100%) and $\theta_w$:
wilting point (100 %); $\bar{X}$: mean value of the five soil layers; SD: standard deviation of the five soil layers.



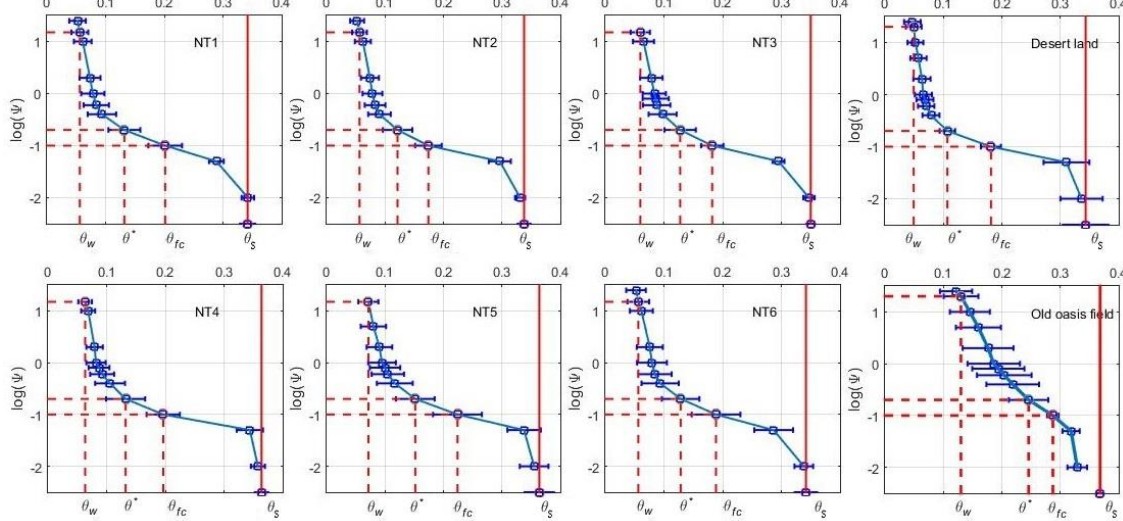


**Figure 4.** *Soil moisture characteristic curve (SMC) of uniform soil profiles of the six experimental plots and two other representative fields. Soil field capacity ($\theta_{fc}$), wilting point ($\theta_w$), and water stress point, i.e., point of incipient stomatal closure ($\theta^*$) are empirically related to the corresponding soil matric potentials (-0.1 bar for $S_{fc}$, -0.2 bar for $\theta^*$ and -15 bar for $S_w$); the blue horizontal line represents the error bar, and the solid red line represents saturated water content ($\theta_s$), which was obtained via the traditional soil drying method with 3 repetitions in each layer; for soil water (matric) potential ($\Psi$) take the absolute value, for example, -0.01 bar is equal to -2 on the Y axis.*

## 3.2 Meteorological and irrigation data

The mean temperature of the growing season in 2016 was 27.12°C, or 3.12 degrees Celsius warmer than the long-term average of the growing seasons in 2007-2016 (24.0°C), and the mean rainfall during the period was about 60.2 mm, or 47 percent less than the long-term average of 115.4 mm (2005-2016), indicating that the weather was hotter and drier during the growing season in 2016 than in the previous ten years. Fig. 7 presents a summary of the amount of water applied over the entire growing season of 2016. Irrigation applications began in mid-April and continued until late September, every 5 to 25 days, depending upon moisture content and crop growth (Fig. 3). A total of 10 irrigation events were sequentially applied through furrow irrigation for the plot during the entire growing season. Based on the in-situ observations of irrigation—i.e., the power consumption of the pumping irrigation well—the estimated irrigation volumes of the six plots were averaged and tested against the observations at field scale. The estimated average cumulative irrigation volume of the six plots during the entire growing season was 831.6 mm (i.e., 1187, 760, 652, 840, 683, and 867 mm for NT1 to NT6, respectively), which compares well with the actual average irrigation volume (868.8 mm) determined through power consumption, suggesting that the calculated irrigation agrees closely with the real values from the farm fields when accurate irrigation and rainfall data are available. A difference of 4.5% in the irrigation amount was observed between the real values and the estimated values over the entire growing season of 2016, indicating a high reliability of the water balance method used in the *SWBCs* estimation.

## 3.3 Soil moisture dynamics (SMDs)

Because the inverse method proposed by Zuo et al. (2002) and Guderle and Hildebrandt (2015) had never been applied throughout an entire growing season for farmland, checking the soil water dynamic of the entire growing season can help us verify the boundary setting and affirm the assumption about the irrigation estimation used. Fig. 2a shows an example of the soil water content responses at various depths of NT6 during and after the irrigation event of 107.1 mm on DOY 154 (2016). TDR measurements exhibited a sharp increase when irrigation began and then decreased rapidly as it was turned off, due to the poor water-holding capacity of the sandy soil. The increase in water content occurred layer by layer from the upper horizons, suggesting limited influence from potential preferential flow (Liu and Lin, 2015), while the rapid moistening of the deep horizons could imply the existence of water loss by drainage. The greatest rate decrease in water content was observed in the top 20 cm of soil. During the 12 h after irrigation, the water content at the top sensor decreased from 21.9% to 14.2%. For the same interval of time the water



contents in the 40-, 60-, 80- and 100-cm depths of soil decreased from 25.4%, 19.8%, 18.5% and 14.2% to 15.7%, 14.3%, 15.4%
and 12.8%, respectively. After irrigation ended, water continued to move down the soil profile; and thus the top part of the profile
was continuously losing water to the soil below it. The lower soil horizons were leaching water into the horizon below but at the
same time were receiving water that had drained from the horizon immediately above, resulting in lower rates of decrease in water
content for these layers than for those at the top horizon (20 cm) (Fares and Alva, 2000). Very similar patterns of changes in water
content were observed through the six different soil profiles.
The average field capacity value ($\theta_{fc}$) of NT1-6 determined from laboratory measurement of soil water release curves was
19.2% (i.e., 20%, 17%, 18%, 19%, 22% and 19% for NT1-6 respectively). Twenty-four hours after the end of irrigation (June 3,
2016), the soil moisture values for the all the measured horizons (20-100 cm depth) of NT1-6 ranged between 8.9% and 16.9%
(13.7-15.7, 13.7-15.1, 8.9-14.5, 9.6-16.9, 11.7-15.3 and 12.3-14.2% for NT1-6 respectively), lower than the field capacity
(Figs. 2 and 5), suggesting that the rapid drainage of water away from the root zone soil (0-100 cm) was terminated during the
period, as expected. In the mornings of the subsequent days, the decrease in soil moisture again sped up as the evaporative demand
of the atmosphere gradually increased. In the absence of any irrigation during the subsequent nights, a slow-down in the decrease
or even a very light increase, in the soil moisture content was observed in the top soil layer (Fig 2). According to the data, there was
also no obvious response of soil moisture regimes to precipitation, indicating a very limited contribution of rainfall to the soil water
storage compared with irrigation. In fact, more than 90% of the rainfall events in this region are less than 5 mm (Fig. 3), and canopy
interception (about 2-5 mm) may have hampered any effective infiltration from those insufficient precipitation events.

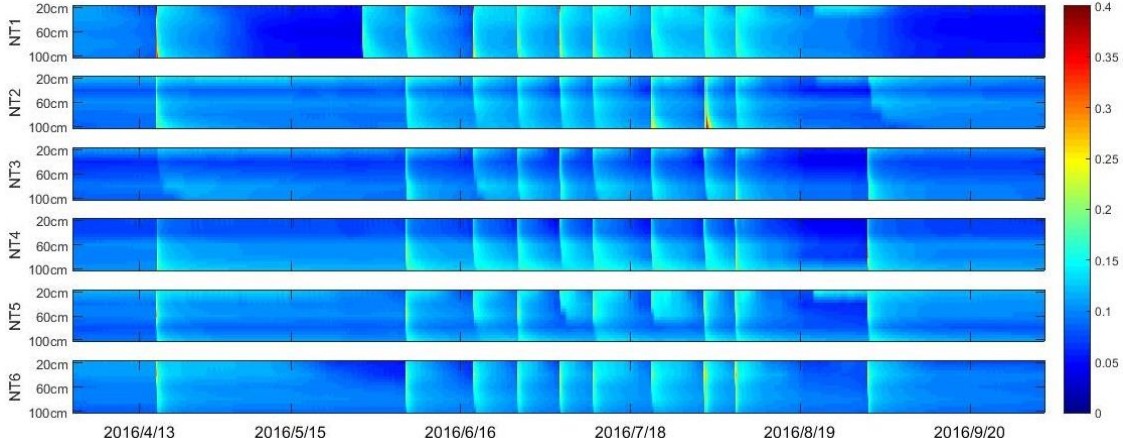

*Figure 5.Spatial and temporal variations of soil water content with a time resolution of ten minutes. The color bar on the right side represents*
*volumetric soil water content. Time period was from Apr. 1 to Oct. 1, 2016. Irrigation events for NT2-6 occurred on 4/16, 6/2, 6/15, 6/23, 7/1, 7/7,*
*7/18, 7/28, 8/3, and 8/28. NT1 had one more irrigation event on 5/25 and one less on 8/28.*

### 3.4 Soil water budget components (*SWBCs*)

The estimated soil water budget components, including total irrigation, deep percolation, and *ET*, at the six different plots
during the growing season of 2016 are summarized in Table 3 and Fig. 7. Evapotranspiration and deep percolation dominated the
fields' relatively simple soil water budgets during the study period. A clear trend in seasonal variation of the water budget
components can be observed at the site (Fig. 7). The corresponding ET values were very similar for all the plots. Three different
stages of ET could be discriminated throughout the 2016 growing season: ET rate was very low at the initial stage (i.e., the first 50
days of the growing season), and increased gradually as vegetation coverage became greater with crop development, before reaching
maximal values at the mid-season stage. After that, ET decreased gradually until harvest time. The estimated daily ET values ranged
largely between 0.2 and 12 mm d$^{-1}$, with an average of 3 mm d$^{-1}$. No significant differences were detected in the daily ET when
Duncan's multiple range test was applied at the 5% level to compare among the six experimental plots (*P*>0.75). A relatively large
difference was observed between selected plots in this study, i.e., significantly higher cumulative irrigation volume was found at
NT1. The excess of water in the soil produced an important deep percolation, which became greater with the increase in the irrigation





357 quota. Among the plots, 45-79% of the input irrigation water was consumed by way of ET (i.e. for plant growth), while the change

358 in soil water storage before and after the growing season was quite small. It is clear that although there was a high correlation

359 between the volume of irrigation and that of drained water, the superfluous irrigation amount had limited influence on the

360 accumulated ET during the growing season.


362 **Table 3.** *Estimated evapotranspiration and other major soil water budget components during the growing season of 2016*

| Cumulative SWBCs | NT1 | NT2 | NT3 | NT4 | NT5 | NT6 |
|---|---|---|---|---|---|---|
| Irrigation | 1186.5 | 760.1 | 652.2 | 840.4 | 683.2 | 867.3 |
| Drainage | 651.8 | 288.3 | 170.7 | 340.1 | 212.4 | 364.7 |
| ET | 534.6 | 489.1 | 508.8 | 561.9 | 539.2 | 538.1 |
| Storage diff.* | -52.7 | 0.17 | 3.6 | 2.2 | 5.44 | -11.64 |

363 *\* Storage differences represent the difference in soil water storage before and after the growing season.*

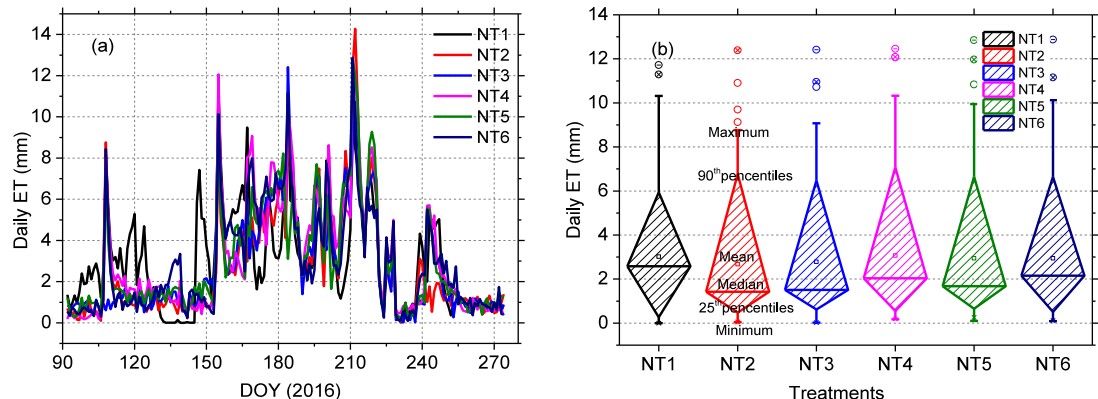

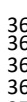

367 **Figure 6.** *Daily ET during the growing season of 2016 as determined from the inverse Richards method: a) time series of estimated daily ET; b)*
368 *box-and-whisker diagrams showing the minimum, median, 25th percentile, 75th percentile, and maximum daily ET. No significant differences were*
369 *detected when Duncan's multiple range test was applied at the 5% level to compare values among the plots.*

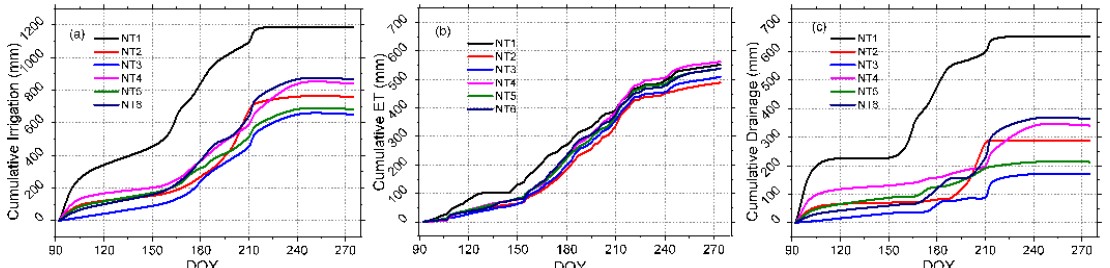

372 **Figure 7.** *Estimated water components of the plots during the growing season of 2016: a) cumulative irrigation, b) cumulative ET, c) cumulative*
373 *drainage*

## 375 4. Discussion

### 376 4.1 Estimated ET

377  Cumulative ET values calculated from inverse Richards methods ranged between 489.1 and 561.9 mm for the different

378 treatments in 2016. The values of ET obtained from the current study are well within the range of published ET values at the nearby

379 sites (406-778 mm), and are consistent with the averages from other studies (~585.5mm) also done in this region, including Zhao

380 and Ji (2010); Rong (2012); Yang et al. (2015); You et al. (2015); Zhao et al. (2015), etc. for maize fields similar to the ones present

381 at the study site (Table 4). Compared with the methods used in the literatures listed in Table 4, the soil-moisture data-based method

382 used in this study is more reliable because it produced a better fit between the numerical solution (soil water profile calculated by

383 the inverse Richards equation) and the measured values of soil moisture content (soil water profile measured by TDR), even with



vertical flow accounted for (Guderle and Hildebrandt, 2015). The narrow range of cumulative ET (489.1-561.9 mm) observed in
2016 can be attributed to the similar sandy soil texture and mesic moisture regimes caused by frequent irrigation (Figs. 4 and 5),
which in turn suggested that for the unmulched alfalfa and mulched maize, both cropping systems and agronomic manipulation had
limited influence on the accumulated ET during the growing season (Srivastava et al., 2017). This result is well supported by the
evidence reported by early investigators, that the ET differences in different cropping systems are quite small for coarse-textured
soils compared with the large differences in the amount of irrigation water (Jalota and Arora, 2002;Ji et al., 2007), and that ET is
strictly a function of ambient atmospheric conditions under normal or wet conditions (Rahgozar et al., 2012).
The observed seasonal trend of ET corresponded well to the irrigation frequency and crop water consumption characteristics
of the growth stage (Fig. 7), and similar patterns in the ET processes have also been reported by many other researches conducted
in this region (Zhao et al., 2015;Zhao et al., 2010). Although we also noticed that the cumulative ET of NT1 was relatively higher
than those of the other plots at the beginning of the growing season, this phenomenon can be largely attributed to the plastic film
mulching at the other five plots. In the early growing season (seeding to emergence), soil evaporation (E) is the major part of ET
(Zhao et al., 2015), and the plastic film mulching applied to NT2 to NT6 was able to significantly retain the soil moisture and thus
decrease soil evaporation (Jia et al., 2006). However, the differences in the cumulative ET, between NT1 and the other plots, were
quite small after the mid-growing season, most likely because with the plant canopy development, crop transpiration became the
major portion of ET, and the influence of plastic film on ET diminished (Zhang et al., 2017;Qin et al., 2014;Jia et al., 2006). Another
influence that may have decreased the evapotranspiration at NT1 after the mid-growing season is cutting. Cutting alfalfa lowers the
leaf area index (LAI) and drastically changes the effective diffusive resistance, consequently lowering the daily ET rate of alfalfa at
NT1, although for a short time after cutting, evaporation from the soil surface may compensate for the decrease in transpiration
(Dong et al., 2003;Su et al., 2010).
*Table 4. Reported ET of oasis maize field in the middle Heihe River Basin (HRB)*

| ET (mm) | Growing period | Year | Soil type | Irrigation | Rainfall | Methods | Paper |
|---|---|---|---|---|---|---|---|
| 651.6 | Apr.11-Sep.18 | 2001 | --- | 690 | 84.4 | Water balance methods | (Peixi et al., 2002) |
| 513.2 | Apr.16-Sep.22 | 2005 | Light loam | 360 | 153.5 | Bowen ratio method | (Jinkui et al., 2007) |
| 486.2 | Apr.16-Sep.22 | 2005 | Light loam | 360 | 153.5 | Reference ET-crop coefficient method | (Jinkui et al., 2007) |
| 777.75 | Apr.21-Sep.15 | 2007 | Sandy loam | 1194 | 102.1 | Bowen ratio method | (Zhao et al., 2010) |
| 693.13 | Apr.21-Sep.15 | 2007 | Sandy loam | 1194 | 102.1 | Penman | (Zhao et al., 2010) |
| 618.34 | Apr.21-Sep.15 | 2007 | Sandy loam | 1194 | 102.1 | Penman-Monteith | (Zhao et al., 2010) |
| 615.67 | Apr.21-Sep.15 | 2007 | Sandy loam | 1194 | 102.1 | Water balance method | (Zhao et al., 2010) |
| 560.31 | Apr.21-Sep.15 | 2007 | Sandy loam | 1194 | 102.1 | Priestley-Taylor | (Zhao et al., 2010) |
| 552.07 | Apr.21-Sep.15 | 2007 | Sandy loam | 1194 | 102.1 | Hargreaves method | (Zhao et al., 2010) |
| 671.2 | Apr.10-Sep.20 | 2009 | Sandy loam | 797 | 97.7 | FAO-56-PM and dual crop coefficient method | (Zhao and Ji, 2010) |
| 640 | Apr.10-Sep.20 | 2009 | --- | 797 | 97.7 | Shuttleworth-Wallace dual-source model | (Zhao et al., 2015) |
| 570—607 | Apr.22-Sep.23 | 2010 | Loamy sand | 990-1103 | 75 | Field experiments | (Rong, 2012) |
| 405.5 | Apr.20-Sep.22 | 2012 | Clay loam | 553 | 95.9 | Water balance and isotope methods | (Yang et al., 2015) |
| 450.7 | Apr.20-Sep.22 | 2012 | --- | 430 | 104.9 | Eddy covariance system | (You et al., 2015) |
| 554.0 | Apr.20-Sep.22 | 2012 | --- | 430 | 104.9 | Penman | (You et al., 2015) |
| 489-562 | Apr.10-Sep.20 | 2016 | Sandy soil | 652-867 | 60.2 | Inverse method | This paper |


**4.2 Other estimated *SWBCs* in this study**
The irrigation volume of maize (NT2 to NT6) within our plots ranged between 652.2 and 867.3 mm, with an average value of
760.6 mm, which is well comparable to the range of average maize field irrigation volume in this region, i.e., a range between 604.8
and 811.4 mm reported in the Statistical Yearbook of Zhangye City for the period of 1995 to 2017 (see http://www.zhangye.gov.cn).
When compared to the other treatments of plastic film mulching, significantly higher amounts of the applied irrigation (1186.5 mm)
were found in NT1, which could be attributed to the larger percentage of infiltrating surface area and the relatively longer irrigation
duration caused by rougher surface of the ground without plastic film mulching. According to Yang et al. (2018a), plastic film mulch
has been widely used to increase the productivity of crops in arid or semiarid regions of China. The logic behind this approach is
that plastic film mulch improves the soil physical properties, such as the soil water content and temperature in the top soil layers,
and thus leads to increased plant growth and yield (N. Mbah et al., 2010). Our results suggested that plastic film mulching can
equally reduce irrigation duration and applied water depth by lowering surface roughness and thus the friction coefficient of the
ground. Similar results were also reported by earlier investigators (Zhang et al., 2017;Jia et al., 2006;Qin et al., 2014).
A less extreme but still significant difference can be found in the irrigation volumes (~652.2 to 867.3 mm) over the other five plots
with plastic film mulching (NT2-6). This may be associated with the inconsistent durations caused by uneven irrigation applications,





randomly rough soil surfaces, and mutation of the infiltration rate (i.e., $K_s$) across the plots (Table 2). Uneven irrigation may be
further attributed to the uneven fields and ditches, which may lead to the application of much more water than required for
evapotranspiration, in some places (Babcock and Blackmer, 1992). Soil surface texture has a direct effect on soil water and complex
interactions with other environmental factors (Yong et al., 2014). The hydraulic behavior and the rate of traditional surface irrigation
is eventually influenced by the inflow and duration of each irrigation (Ascough and Kiker, 2002). Although only slight differences
exist among the retention curves (Fig. 4), the differences in saturation water conductivity ($K_s$) can be substantial (varying between
119 cm/day at NT1 and 286 cm/day at NT3), indicating that a slight difference in hydrophysical properties of soil profiles could be
amplified to generate wildly varying infiltration behavior, especially during saturated or near-saturated stages under actual irrigation
conditions (Ojha et al., 2017).
In desert oasis farmland, the water cycle is primarily driven by evapotranspiration demand under the influence of irrigation,
and soil water percolation may occur when too much water is applied to the root zone. Estimated deep drainage rates were observed,
ranging from 170.7 mm (NT3) to 651.8 mm (NT1), amounting to about 26.2% and 54.9% of the total irrigation of the two plots,
respectively. Drainage within the mulched maize fields ranged from 170.7 mm to 364.7 mm, which are in good agreement with
other results from the same region, i.e., 255 mm through isotopes obtained by Yang et al. (2015), and 339.5 mm through the Hydrus-
1D model by Dong-Sheng et al. (2015). Compared with the theoretical deep drainage determined by water balance techniques (Rice
et al., 1986), an error of -2.6 to 43.1 mm, or 0.2 % to 17.6%, was obtained for the cumulative deep drainage (Table 3), indicating
the reliability of the method used to estimate deep drainage in this study. The data expressed in Fig. 2 also explain how easily an
excess of water, and therefore deep drainage, can occur in these soils. Indeed, the deep drainage was directly proportional to the
amount of irrigation applied during any particular period (Fig. 7, Table 3). This phenomenon is easy to understand because for a
given amount of irrigation, the likelihood of a drainage event and its average size both increased naturally with the irrigation amount
(Fig. 7) (Keller, 2005). It is obvious that drainage should be an essential part of irrigation design and management. According to our
results, an average of 40.6% of input water was consumed by deep leakage across the six plots; this is unproductive and could even
cause nutrient loss and groundwater pollution at field scales (Fares and Alva, 2000), suggesting there is a huge potential for
increasing irrigation water-use efficiencies and reducing irrigation water requirements in this region.

### 4.3 Effects of different cropping systems and tillage periods on soil hydrophysical properties

In this desert oasis with constant expansion, most of the fields belong to smallholder farmers, who usually follow different
tillage periods and special cropping patterns, resulting in a heterogeneity of soil hydrophysical properties (Salem et al., 2015;Ács,
2005;Abu and Abubakar, 2013). For the soil-moisture data-based method proposed in this paper, the spatial heterogeneity of the soil
hydrophysical properties—which can be characterized by hydrophysical functions (soil water retention curve and soil water
conductivity) and/or hydrophysical parameters ($\rho_b$, $\theta_s$, $\theta_{fc}$ and $\theta_w$) (Ács, 2005)—may restrict its applicability to a large
agricultural area. Therefore, evaluating to what extent the different cropping systems and agronomic manipulations affect the soil
hydrophysical properties is important, in order to reduce unnecessary repetitive measurements of soil hydrophysical information at
both spatial and temporal scales, and thus improve the application efficiency of our method. Long-term cropping can increase annual
water productivity by improving soil hydrophysical properties and reducing unproductive water losses (Caviglia et al., 2013). Crop
root systems, for example, may create heterogeneity in soil properties through mechanical actions and the active release of chemicals
(Hirobe et al., 2001;Read et al., 2003); and, along with similar feedbacks between long-term planted crops and the soil environment,
may change water flow and soil hydraulic characteristics, and thus affect local water balances (Baldocchi et al., 2004;Séré et al.,
2012). Although it is difficult to quantify the consequences of plant-soil feedbacks on the hydrologic cycle of farmland, because of
the lack of an accurate simulation model (Jalota and Arora, 2002), our results indicated that the tillage and planting of past decades
have significantly increased the soil's water-holding ability (i.e., higher values of $\rho_b$, $\theta_s$, $\theta_{fc}$ and $\theta_w$ compared with the sandier
land). The magnitude of increase in most of the parameters, except $K_s$ in soil vertical profiles, was independent of the treatments
applied across the six selected plots, which also suggests that different cropping systems and agronomic manipulation have limited
effects on differing soil physical characteristics in sandy soil, at least at a decade scale, and this agrees well with the reports from
Katsvairo et al. (2002). However, we argue that significant differences in soil hydrophysical properties among the plots may occur
if the treatments are conducted over longer periods of time, i.e., ~100 years or more. In summary, the relatively slow process of soil
evolution with tillage operations, and the limited influence of different cropping systems on soil hydrophysical properties at a 10-





year scale, indicate a good stability and representativeness of the measured soil hydrophysical data and thus a good application
prospect for applying the soil-moisture data-based method in practice.

### 4.4 Potential for *SWBC* estimation by using soil moisture measurements

The best estimates of *SWBCs* should be based on models of soil water, because in most cases direct measurements are not
available (Campbell and Diaz, 1988). Many studies including modeling work have been conducted in this region during the past
decades (Table 4). Since there has been a lack of accurate parameters to assess the heterogeneity and complexity involved in
modeling (Allen et al., 2011;Suleiman and Hoogenboom, 2007;Wang and Dickinson, 2012;Ibrom et al., 2007), however, most of
these were rough approximations based on meteorological methods and water balance equations (Rong, 2012;Jiang et al., 2016;Yang
et al., 2015;Wu et al., 2015;Ji et al., 2007). Yet soil-moisture data-based methods have been considered one of the most promising
ways to directly determine ET and other *SWBCs* (Guderle and Hildebrandt, 2015;Li et al., 2002), and many possible options,
including single- or multi-step, and single- or multi-layer water balance methods, have been proposed and tested with synthetic time
series of water content (Guderle and Hildebrandt, 2015). Our results suggest that a combination of a soil water balance method and
the inverse method could be a good candidate for SWBC estimation in this region, and can provide a reliable solution, especially in
regards to estimating ET, root water uptake, and water vertical flow, and do not require any prior information of root distribution
parameters, and they can be applicable under both wet and dry weather conditions (Guderle and Hildebrandt, 2015).
Information on *SWBCs* is crucial for irrigation planning at both the field and regional scale (Jalota and Arora, 2002). Early
researches suggested that decreasing the irrigation amount and increasing the irrigation frequency is the best choice for saving water
and improving water use efficiency in the middle HRB (Rong, 2012;Jiang et al., 2016;Yang et al., 2015;Wu et al., 2015;Ji et al.,
2007). This scenario can be achieved not only by adopting proper modern irrigation systems but also by integrating new technologies
into the effective planning of irrigation schedules, so that plants can be supplied with optimal water volume and minimum water
loss. Soil water budget models help in translating irrigation amounts in different time periods to evapotranspiration (ET), which has
significance from the standpoint of crop yield (Jalota and Arora, 2002). Our results show that superfluous irrigation has no effect on
increasing ET, because of the poor water-holding capacity of the sandy soil in this region, and thus irrigation application should not
exceed a specific threshold (i.e., root zone depletion, ~527 mm for maize) to avoid deep percolation, which has a negative effect,
increasing irrigation costs (Zotarelli et al., 2016). However, water deficits in crops and the resulting water stress on plants also
influences crop evapotranspiration and crop yield (Kallitsari et al., 2011). Thus, a soil moisture measurement method based on
SWBC estimation makes it possible to quantify water budget components for different time periods, and has great potential for
identifying appropriate irrigation amounts and frequencies. As the price of commercial TDR systems has become affordable
(Quinones and Ruelle, 2001), it is more and more frequently used for soil water content measurements in desert oases, and thus a
soil-moisture data-based method has great potential in irrigation management optimization and in moving toward sustainable water
resources management, even under traditional surface irrigation conditions (Tawara *et al.*, 2015).

### 4.5 Uncertainty analysis

Uncertainty is inevitable, in any soil water budget components estimate. As summarized by Zuo et al. (2002) and (Guderle and
Hildebrandt, 2015), the accuracy and convergence of estimated evapotranspiration and slow drainage using this inverse approach
are dependent on several factors, including the accuracy of soil hydraulic parameters and input soil moisture data, the time intervals
of soil water content measurements, the spatial interval of the measured data along the depth, the setting of simulation depth and
the boundary conditions. For a soil-moisture data-based method, the estimated results are only as good as their input data, i.e., the
accuracy, the precision and the resolution (Guderle and Hildebrandt, 2013;Guderle and Hildebrandt, 2015). In this study, every effort
was made to eliminate the uncertainty caused by the quality of the input data: for example, all the sensors and cables were carefully
buried according the operator's manual instructions; the soil-specific calibration of TDR was conducted in a well-designed
laboratory calibration experiment, which results a good accuracy ($\pm 2$ %) for TDR measurement in coarse-textured soil; and the
high-resolution moisture data (taken at 10-minute intervals) were hourly averaged to numerically filter out the noise and improve
the calculation speed of the inverse model. Meanwhile, the simulation depth (0-110cm) is consistent with the root depth, and it can
be well represented by 5 TDR probes with a spatial interval of 20 cm in sandy soil (Zhao et al., 2016). The boundary condition is
also important for this inverse model (Liao et al., 2016); as mentioned in Section 2.3.3, we set the upper and lower boundaries as





close as possible to natural conditions. However, we did not set specific upper boundaries for inter-cropping treatments, i.e., no bare soil evaporation was considered in the inter-cropping maize-pea field, which may have slightly underestimated the ET of NT6, but within an acceptable range because the soil evaporation of NT6 was relatively small when compared with the total transpiration over a growing season. Moreover, the high amount of irrigation may have reduced the temperature of the soil profile, because irrigation is often accompanied by an increase in latent heat flux, and thus by an increase in evapotranspiration (Chen et al., 2018;Haddeland et al., 2006;Zou et al., 2017). Theoretically, a decrease in soil temperature may slightly increase the soil suction under the same moisture conditions (Bachmann et al., 2002), and hence variations in the soil temperature profile under different irrigation scenarios may have affected the accuracy of the inverse model by changing the soil water retention curves. However, irrigation-affected variations of soil profile temperature in this study were small (within 2℃), which is smaller than the daily variation of soil temperature (2 to 3℃), and thus its effect on soil water retention curves can be ignored for eco-hydrological researches (Bachmann et al., 2002;Gao and Shao, 2015). Even so, it is still an interesting and important research field deserving further investigation.

Aside from the uncertainties in estimating evapotranspiration and slow drainages, more limitations may exist in the estimation of irrigation amounts and rapid drainages following irrigation events. Both of these limitations were strongly dependent on the assumptions of Equation (2) and (3), specifically, the estimation of $S_{max}$. We checked all the irrigation events of NT1-NT6 during the entire 2016 growing season, and results showed an acceptable accuracy of the estimation of $S_{max}$ (only two irrigation events in NT2 slightly underestimated the $S_{max}$: 1.86 and 10.3 mm, which accounted for 1.1% and 4.1% of total soil water storage, respectively). This phenomenon—deep percolation that began before irrigation ceased—may have been caused by long irrigation duration time and high $K_s$ of surface soil at NT2, which is the major limitation when applying our method to other regions. Calculating the previously occurring leakage volume, for example, using the unsaturated hydraulic conductivity empirical equation, is one of the possible solutions that needs to be tested in future work. Installing TDR under the film-mulched ridges may also cause an underestimation of the soil moisture content during an irrigation event. We investigated the difference caused by the location of TDR by comparing the soil water dynamics of an unmulched flat plot (NT1, which was independent of TDR location) and film-mulched ridge plots (NT2-6, which were affected by TDR location) after irrigation, and found that the underestimation caused by the location of TDR was mainly significant in the top 30 cm of the soil layer. For example, during the 24 hours after the irrigation on June 2 (DOY 154-155, Fig. 2), in the top 30 cm of the soil layer, the maximum soil moisture value of NT1 was 0.378 while the maximum soil moisture value of other plots (NT2-6) ranged between 0.219 and 0.299; in other layers, the maximum soil moisture value of NT1 was well within the maximum soil moisture values of other plots at the same layer, i.e., 0.189, 0.191, 0.174, 0.164 for NT1 and 0.154-0.254, 0.153-0.277, 0.154-0.205, 0.148-0.181 for the other plots. The minimum soil moisture values were very close between NT1 and the other plots at the same layer (<0.04). Meanwhile, the variances between NT1 and the other plots were 0.006 to 0.009 in the top 30 cm of the soil layer, and 0.001-0.004, 0.003-0.004, 0.001-0.003, 0.002-0.004 for the other layers, which showed a good consistency of soil dynamics in the 30- to 110-cm soil layers compared with the top 30 cm of the soil layers. These consistencies may be because by 1) the height of ridge shoulders in the experimental plots was relatively low (<3cm), and substantial infiltration could occur through the film holes made for maize growth; 2) lateral water transfers could be substantially enhanced during the period of irrigation because of the soil water potential differences between ridges and furrows. This judgment also can be supported by some researches conducted in similar environments, e.g., Zhang et al. (2016). Therefore, we argue here that the uncertainty that TDR location brings to the *SWBC* estimations in this study is acceptable. For now, given that the effect of plastic mulched furrow irrigation on soil water distribution remains elusive (Zhang et al., 2016;Abbasi et al., 2004), installing TDR in both the ridge and the furrow may be a better choice in future studies. Besides, both the heterogeneity of soil hydrophysical properties in sandy soils and the rough artificial irrigation process can bring uncertainties in the irrigation amount of any oasis cropland. However, the maximum irrigation rate of flood or furrow irrigation is mainly dependent on the $K_s$ of the top soil layer, which is nearly homogeneous in such small experimental plots (6m×9m) because they have the same cropping systems and agronomic history (Table 2), and thus there is no significant infiltration difference within one small plot, and the installed soil moisture probes can well monitor the irrigation process of the entire plot.

Overall, we are confident about the estimation accuracy of ET, which is the most important parameter among all the *SWBCs*, and the one the related researchers are most interested in, because of its direct relevance to crop yield, and because maximizing crop yield is the major objective of agricultural irrigation strategies (Liu et al., 2002;Zhang et al., 2004;Kang et al., 2002). The ET





estimation model in this study not only has great advantages in theory (for example, it does not require any root distribution
information (Schneider et al., 2010;Guderle and Hildebrandt, 2015)), but at the same time it also considers the hysteresis effect,
unlike other common models (Li et al., 2002;Guderle and Hildebrandt, 2015), while also providing a reliable and high-resolution
solution because its results are well within the range of published ET values at nearby sites. Other *SWBC* estimations such as
irrigation, also had an acceptable accuracy, even though they were estimated by a relatively simple method, because the results show
a good consistency with the observations (actual irrigation calculated from the power consumption) at the field scale and with the
average irrigation amounts in other maize fields in the same region at close to the same time.

## 5. Conclusions

A database of soil moisture measurements taken in 2016 from six experimental fields (which were originally designed to test
the accumulative impacts of different cropping systems and agronomic manipulations on soil-property evolution in the ecotone of
desert and oasis) in the middle Heihe River Basin of China, was used to test the potential of a soil-moisture time series for estimating
the *SWBCs*. We compared the hydrophysical properties of the soils in the plots, and then determined evapotranspiration and other
*SWBCs* through a soil-moisture data-based method that combined both the soil water balance method and the inverse Richards
equation, and the uncertainties of the employed methods were analyzed at the end of the experiment. Our results confirmed that (1)
relatively reasonable estimations of the *SWBCs* in a desert oasis environment can be derived by using soil moisture measurements.
Although uncertainties exist, our method, which balanced simplicity and accuracy, can provide a reliable solution, especially in
regards to estimating ET, for coarse-textured sandy soils; (2) although the tillage and planting of the past decade have significantly
increased the soil water-holding ability, the magnitude of increase in most of the soil hydrophysical parameters was independent of
the different treatments applied across the plots during a 10-year period, resulting in a good prospect for applying our method among
different fields; (3) the estimated results of the *SWBCs* will provide a valuable reference for optimizing irrigation strategies at the
filed scale, but it is still a long way from use on large areas of agricultural land, because of the soil heterogeneity at the regional
scale and the small volume that a TDR probe can monitor.

## Acknowledgements

This research was jointly supported by the National Natural Science Foundation of China (No. 41630861), the Youth
Innovation Promotion Association of Chinese Academy of Sciences, and the West Light Foundation of Chinese Academy of
Sciences (No. 29Y929621). We would like to thank Dr. Yang Yu for his constructive suggestions on completing this work. Special
thanks also go to editor Fuqiang Tian, Dr. Jun Niu, Dr. Yanjun Shen, Dr. L. Brocca, and the other anonymous reviewers, whose
perceptive criticisms, comments and suggestions helped us improve the quality of the manuscript.

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
