# Peer review of "Quantification of Soil Water Balance Components Based on Continuous Soil Moisture Measurement and Richards Equation in an Irrigated Agricultural Field of a Desert Oasis"

_Hydrology and Earth System Sciences, 2019_

## Referee Comment (RC1) · Anonymous Referee #1 · 23 Apr 2019

The manuscript uses timeseris of soil moisture data to estimate the soil water budget components, especially evapotranspiration, in an irrigated agricultural filed of a desert oasis. This study is well conducted, and the authors responded the comments well. I think it is worth publishing after minor revisions. I have several comments listed as below.

1. The anonymous reviewer 3# concerns the locations of Secion 3.1, 3.2, and 3.3, and the authors insisted the former locations. I think the problem may be arisen by the title of them. The current titles may be misleading to about the dataset, nor the observed

or calculated results. For example, Section 3.2 is about calculated irrigation amount. The meteorological data should be introduced in the materials section.

2. I am confused by the Sstop and Smax. Is Sstop larger than Smax, as shown in Fig. 2.

3. Please check the captions appeared in the text. For example, Fig.7 appears early than Fig. 5. Where is Fig. 6?

4. Fig.3, It is better to shown the specific irrigation amount and compare it with rainfall event.

---

## Short Comment (SC1) · 8 May 2019

*A note upfront from the submitting person: This review was prepared by Basil Frefel and Michèle Bösiger, both master students in geography at the University of Zurich. The review was part of an exercise during a second semester master level seminar on "the biogeochemistry of plant-soil systems in a changing world", which I organize. We would like to highlight that the depth of scientific knowledge and technical understanding of these reviewers represents that of master students. We enjoyed discussing the manuscript in the seminar, and hope that our comments will be helpful for the authors.*

[Figure]

Addressing sustainable irrigation in semi-desert regions, Li et al. observed in a soil-moisture time series the soil-water holding ability in the Heihe River Basin, northern China. Soil properties such as saturated hydraulic conductivity and soil retention capabilities were combined with the soil water balance method and the inverse Richards equation (water movement in unsaturated soil) in order to estimate the Soil Water Base Components (evapotranspiration, irrigation, drainage). The measurements were taken at six sites in sandy soils, which differed in agricultural technique (rotational, permanent cultivation), plant species and mulching application. The results show that the estimation of the Soil Water Base Components corresponds to the soil-moisture time series and thus should be helpful for future irrigation planning.

Overall, the issue addressed by Li et al. is of great importance regarding future water management. Especially with increasing water scarcity, the study lays the foundation for a sustainable water strategy for one of the biggest agricultural producers worldwide for rice and wheat and thus is a valid contribution to the present scientific knowledge. The article is well structured, and the thinking steps are described profoundly, in addition to the reflected approach to the results. However, we have also found some caveats. In general, the type size is too small, and the sentences tend to be too long and interlaced to comprehend the article at the first read (see further comments). Moreover, the exact time span of the experiment is unclear, and it should be mentioned what (important) role sandy soils play in the agricultural production of China and the world. Besides, the abstract should be shortened to stay attractive for the reader and abbreviations such as NT1-NT6 should not be used in this very first paragraph. In our opinion, many explanations are too complicated; Keep explanations short and simple. In contrast to the detailed abstract, the conclusion is too short, again overloaded with abbreviations and therefore not understandable standing alone. More detailed remarks are listed below.

Unclear section, parts which need further specification: 12: SWBC: In this paper the term soil water budget components and its abbreviation SWBC is used as if it were

[Figure]

a standard term in soil science or hydrology. is this really the case? Otherwise it should be mentioned that this term as such is only used in this paper. 22: Since the inverse Richards equation is of great importance in this work, but is not necessarily known to the general public, a brief description of what this method is used for, e.g. in parentheses, would be helpful 26: Why only one site without film-mulch? Comparison to other sites without mulching would have been helpful 29: What should be special about an obvious correlation between the volume of irrigation and drained water? is this really a significant result of the study or could this statement also be omitted? 47: What is a high leaching fraction? Explain. 40-54 and 55-69: The second paragraph is redundant. 106-107: Reference is needed for the sentence "The annual average precipitation..." 107: What does a dryness index of 15.9 mean. Please put this number into context. Is this a high or a low value compared to the surrounding region or the rest of China? Is the dryness Index a common value which is need to be stated? 111: What exactly is meant with sandy soil? What official soil name does it correspond to? 111: leave out "...coarse texture and...." 111: Scotch or Scots pine? Us familiar expression 124-126: Please explain why the different treatments used in the study were chosen. 127: Why using exactly this type of irrigation (furrow irrigation)? Please explain in more detail. 128: Why just using one site with no film-mulching and five with mulching? Not a sufficient comparison possible between the sites 242: Is it not unrealistic to use a ground level of the soil matric potential, even though the water level never reaches that high up? 245: Which software? Please specify. 343: In what dimension is the soil water content measured?

Sentences which were too long or hard to follow: 55-59: The sentence is too long, therefore hard to follow and should be divided into two or three sentences. End the first sentence with a full stop after (Wright, 1971). 92-95: Unorganised sentence order makes it even harder to follow the content. 165-171: Subdivision into subsets probably better for this sentence. 219-226: Hard to follow the derivation

Remarks concerning formal structures (typos, figures, etc.): 42: missing 'the' ...the

Heihe river basin (HRB) is one of the largest... 81: Cross out "...quite common and..." 83: Cross out "...and more..." 85: Also, with this process_, ... 86: ..., almost no work_ have been... 112: ... (planted since the 1970s), include Haloxylon anmmodendron, ... → either no comma or 'including' Figure 1: The figure is not entirely clear→to what do the roots belong on the right?/figure on the left→layout and position of the legend and unprecise placement of the small map of China Figure 2: Probably better 'day of year' as axis label instead of DOY 140: 'was' instead of 'were' 195: 'dominates' instead of 'dominants' Table 1: we propose to insert this nomenclature-table at the beginning of the chapter. Table 2: Vertical lines between wilting point value of one study site and the saturated water conductivity of the next study site could probably increase readability (see attached pdf) Figure 5: Does it need this figure at this place? and what exactly, apart from the clearly visible irrigation events, should be shown with it? Figure 6: left graph→use other scaling, since nothing is readable/ right graph→why is the scale in the middle of the graph (a bit weird position), why using such fancy boxplots if normal rectangular ones could be used? 381: 'Literature' instead of 'literatures'? 437: explains 443: ... (Fares and Alva, 2000), suggesting that there is... In general: Reflect on the placement and use of figures and tables in the work, so that these stylistic tools fulfil their purpose of increasing the attractiveness of a scientific paper.

Further comments: 1-2: as the use of soil moisture measurements is a major part of the scientific work, we would adjust the title as follows: Estimation of Evapotranspiration and Other Soil Water Budget Components, Using Soil Moisture Measurements, in an Irrigated Agricultural Field of a Desert Oasis Or also the following possibility seems easier to understand to us: Estimation of Evapotranspiration, Irrigation and Drainage, Using Soil Moisture Measurements, in an Irrigated Agricultural Field of a Desert Oasis 29-30: Leave out the obvious parts and concentrate on the findings 106: lowest and highest temperatures for winter and summer, respectively→that's logical, no need of repetition. Pleonasm. It would be probably better to use the terms 'minimum' and 'maximum' in this context. 323-324: water content values are difficult to read in the presented form of a listing. 512: Setting upper boundaries would have been a nice

addition 566: It would be desirable for the conclusion to mention what would be appropriate irrigation methods for this variety of agricultural soil.

[Figure]

Basil Frefel and Michèle Bösiger

**Review on Li et al. 2019: Estimation of Evapotranspiration and Other Soil Water Budget Components in an Irrigated Agricultural Field of a Desert Oasis, Using Soil Moisture Measurements**

Addressing sustainable irrigation in semi-desert regions, Li et al. observed in a soil-moisture time series the soil-water holding ability in the Heihe River Basin, northern China. Soil properties such as saturated hydraulic conductivity and soil retention capabilities were combined with the soil water balance method and the inverse Richards equation (water movement in unsaturated soil) in order to estimate the Soil Water Base Components (evapotranspiration, irrigation, drainage). The measurements were taken at six sites in sandy soils, which differed in agricultural technique (rotational, permanent cultivation), plant species and mulching application. The results show that the estimation of the Soil Water Base Components corresponds to the soil-moisture time series and thus should be helpful for future irrigation planning.

Overall, the issue addressed by Li et al. is of great importance regarding future water management. Especially with increasing water scarcity, the study lays the foundation for a sustainable water strategy for one of the biggest agricultural producers worldwide for rice and wheat and thus is a valid contribution to the present scientific knowledge. The article is well structured, and the thinking steps are described profoundly, in addition to the reflected approach to the results. However, we have also found some caveats. In general, the type size is too small, and the sentences tend to be too long and interlaced to comprehend the article at the first read (see further comments). Moreover, the exact time span of the experiment is unclear, and it should be mentioned what (important) role sandy soils play in the agricultural production of China and the world. Besides, the abstract should be shortened to stay attractive for the reader and abbreviations such as NT1-NT6 should not be used in this very first

---

## Referee Comment (RC2) · Anonymous Referee #2 · 26 May 2019

1. Lines 14, 61 and etc.: drainage is not, and has been never considered as a "driver" of hydrological cycle. Irrigation is taken as a factor that interfares hydrological cycle, and seldomly taken as a "driver". You may call evapotranspiration a driver of the hydrological cycle. It is a component of the hydrological cycle as a matter of fact. Usually, the drivers of the hydrological cycle refer to the climatic factors.

2. Lines314, 315. "Because the inverse method proposed by Zuo et al. (2002) and Guderle and Hildebrandt (2015) had never been applied throughout an entire growing season for farmland...", this is hard to say.

[Figure]

3. As indicated in the last paragraph of the introduction section, this work aims to investigate performance of the inverse method to the coarse-textured soils. Thus, "coarse-textured soils" should be focused and highlighted in the discussion. This might have been something new in this paper.

---

## Author Comment (AC1) · 6 Jun 2019

Response to Anonymous Referee #1 (RC1)

General comments

The manuscript uses timeseries of soil moisture data to estimate the soil water budget components, especially evapotranspiration, in an irrigated agricultural filed of a desert oasis. This study is well conducted, and the authors responded the comments well. I think it is worth publishing after minor revisions. I have several comments listed

as below.

RESPONSE: We warmly thank the Anonymous Referee #1 for the overall favorable impression of the work, and for his or her thorough review and the detailed, helpful comments. Please find below your reproduced comments, followed by our point-by-point responses.

Specific comments 1) The anonymous reviewer 3# concerns the locations of Section 3.1, 3.2, and 3.3, and the authors insisted the former locations. I think the problem may be arisen by the title of them. The current titles may be misleading to about the dataset, nor the observed or calculated results. For example, Section 3.2 is about calculated irrigation amount. The meteorological data should be introduced in the materials section.

RESPONSE: We have reorganized the sub-titles of Section 3 as: "3.1 Soil hydrophysical characteristics", "Soil moisture dynamics (SMDs)", and "3.3 Soil water budget components (SWBCs)". The results about irrigation amount has been merged into section "3.3 Soil water budget components (SWBCs)", and the descriptions upon meteorological data has been moved to "2.3 Calculation methods, 3) Boundary setting and data collection". Thanks to the nice suggestion, this part looks much better than before.

2) I am confused by the Sstop and Smax. Is Sstop larger than Smax, as shown in Fig. 2.

RESPONSE: Very nice question, and the answer is: Smax is larger than Sstop. As we mentioned in "2.3 Calculation methods 1) Water storage and irrigation amounts", Smax was defined as the recorded maximum soil water storage of the root zone, and Sstop is the recorded soil water storage when irrigation event ends (moisture of uppermost soil layer starts to decrease). Although the real water storage in root-zone soil should keep constant during the short periods between irrigation ends and deep drainage starts, it is not naturally been recorded by the soil moisture sensors at any time of this period, because of the continuously redistributing soil water profile and limited number of soil

moisture sensor. In this work, the real water storage in root-zone soil was assumed to be equal to Smax, and thus Sstop would tend to approach Smax if more soil moisture sensors were installed in the soil profile. To clarify this point, more detailed explanation of this point will be included in the revision.

3) Please check the captions appeared in the text. For example, Fig.7 appears earlier than Fig. 5. Where is Fig. 6?

RESPONSE: We feel very sorry for the careless. "Fig. 7" appeared in Section 3.2 has been replaced by "Table 3" in the text, and we further checked all the figure captions to avoid any other similar mistakes.

4) Fig.3, It is better to shown the specific irrigation amount and compare it with rainfall event.

RESPONSE: Thanks for the suggestion, Fig.3 will be reorganized to include the average amount of each irrigation event in the coming revision.

---

## Author Comment (AC2) · 6 Jun 2019

Response to Anonymous Referee #2 (RC2)

We thank the anonymous Referee #2 for taking the time to review our manuscript and for their generally positive feedback on our study. Please find below your reproduced comments, followed by our point-by-point responses.

Specific comments 1) Lines 14, 61 and etc.: drainage is not, and has been never considered as a "driver" of hydrological cycle. Irrigation is taken as a factor that interferes

hydrological cycle, and seldomly taken as a "driver". You may call evapotranspiration a driver of the hydrological cycle. It is a component of the hydrological cycle as a matter of fact. Usually, the drivers of the hydrological cycle refer to the climatic factors.

RESPONSE: Thanks for the nice suggestion, we have changed "driven" as "dominated", and changed "driver" as "components". Please see Line 15 (Page 1) and Line 62 (Page 2) in the revision.

2) Lines314, 315. "Because the inverse method proposed by Zuo et al. (2002) and Guderle and Hildebrandt (2015) had never been applied throughout an entire growing season for farmland...", this is hard to say.

RESPONSE: We have removed this sentence in the revision. Please see Line 315 (Page 10) in the revision.

3) As indicated in the last paragraph of the introduction section, this work aims to investigate performance of the inverse method to the coarse-textured soils. Thus, "coarse-textured soils" should be focused and highlighted in the discussion. This might have been something new in this paper.

RESPONSE: Nice suggestion. Some related discussion will be added in the Section 4.2 and 4.5 in the coming revision to highlight the "coarse-textured soils".

---

## Author Comment (AC3) · 7 Jun 2019

**Response to Dr. Michael W. I. Schmidt (SC1)**

**General comments**

*A note upfront from the submitting person:

This review was prepared by Basil Frefel and Michèle Bösiger, both master students in geography at the University of Zurich. The review was part of an exercise during a second semester master level seminar on "the biogeochemistry of plant-soil systems in a changing world", which I organize. We would like to highlight that the depth of scientific knowledge and technical understanding of these reviewers represents that of master students. We enjoyed discussing the manuscript in the seminar, and hope that our comments will be helpful for the authors.*.

Addressing sustainable irrigation in semi-desert regions, Li et al. observed in a soil moisture time series the soil-water holding ability in the Heihe River Basin, northern China. Soil properties such as saturated hydraulic conductivity and soil retention capabilities were combined with the soil water balance method and the inverse Richards equation (water movement in unsaturated soil) in order to estimate the Soil Water Base Components (evapotranspiration, irrigation, drainage). The measurements were taken at six sites in sandy soils, which differed in agricultural technique (rotational, permanent cultivation), plant species and mulching application. The results show that the estimation of the Soil Water Base Components corresponds to the soil-moisture time series and thus should be helpful for future irrigation planning.

Overall, the issue addressed by Li et al. is of great importance regarding future water management. Especially with increasing water scarcity, the study lays the foundation for a sustainable water strategy for one of the biggest agricultural producers worldwide for rice and wheat and thus is a valid contribution to the present scientific knowledge. The article is well structured, and the thinking steps are described profoundly, in addition to the reflected approach to the results. However, we have also found some caveats.

In general, the type size is too small, and the sentences tend to be too long and interlaced to comprehend the article at the first read (see further comments). Moreover, the exact time span of the experiment is unclear, and it should be mentioned what (important) role sandy soils play in the agricultural production of China and the world. Besides, the abstract should be shortened to stay attractive for the reader and abbreviations such as NT1-NT6 should not be used in this very first paragraph. In our opinion, many explanations are too complicated; Keep explanations short and simple. In contrast to the detailed abstract, the conclusion is too short, again overloaded with abbreviations and therefore not understandable standing alone. More detailed remarks are listed below.

**RESPONSE**: We would like to thank Dr. Michael W. I. Schmidt for organizing the seminar to discuss our manuscript. We also warmly thank Basil Frefel, Michèle Bösiger for their thoughtful review, and for the specific suggestions, with which our manuscript is significantly improved in both its clarity and organization. We have taken the time to think through all the review comments and have adequately addressed all the comments item-by-item in the following pages.

**Specific comments**

1) Lines 12: SWBC: In this paper the term soil water budget components and its abbreviation SWBC is used as if it were a standard term in soil science or hydrology. is this really the case? Otherwise it should be mentioned that this

term as such is only used in this paper.

RESPONSE: Thanks for the nice suggestion. No, this abbreviation is not a standard term both in soil science and hydrology, and it is only used in this paper. We have defined that in the Introduction and then to use it in the following text to reduce repeated use of the term and thus to make the paper as concise as possible. According to the suggestion, we have mentioned that "this term as such is only used in this paper" in the revision.

2) Lines 22: Since the inverse Richards equation is of great importance in this work, but is not necessarily known to the general public, a brief description of what this method is used for, e.g. in parentheses, would be helpful.

RESPONSE: Thanks for pointing out this issue. A brief description of what this method is used for have been added in our revision.

3) Lines 26: Why only one site without film-mulch? Comparison to other sites without mulching would have been helpful.

RESPONSE: As we mentioned in Section 2.2, this long-term field experiment was set up in 2007, and the six experiment plots have very different treatments, i.e., from NT1 to NT6, they were sequentially set as: (1) continuous pasture cropping, (2) continuous maize cropping, (3) continuous maize cropping with straw return, (4) maize-maize-pasture rotation, (5) maize-pasture rotation, (6) maize-pasture intercropping. In general, only the maize fields need to be film-mulched, and maize were planted in all the plots except NT1 (which was planted with alfalfa in the growing season of 2016), so that NT1 is the only one plot without film-mulch in this study.

4) Lines 29: What should be special about an obvious correlation between the volume of irrigation and drained water? is this really a significant result of the study or could this statement also be omitted?

RESPONSE: Yes, we do think this is a significant result of the study. Although similar results were also reported by other works, we found that this linear positive correlation is particularly noticeable in the coarse textured soils like the desert oasis in arid China, as we also discussed in Section 4.2. We think this is a useful result for further improve irrigation strategies, and thus would like to keep it here.

5) Lines 47: What is a high leaching fraction? Explain.

RESPONSE: Leaching fraction (LF) represent the ratio of the actual depth of drainage to the depth of irrigation (Dudley et al., 2008), we have explained it in the coming revision. Please see line 48 (Page 1) for details.

6) Lines 40-54 and 55-69: The second paragraph is redundant.

RESPONSE: The second paragraph will be slightly shortened to eliminate any redundant in the coming revision.

7) Lines 106-107: Reference is needed for the sentence "The annual average precipitation..."

RESPONSE: Added as suggested.

8) Lines 107: What does a dryness index of 15.9 mean. Please put this number into context. Is this a high or a low value compared to the surrounding region or the rest of China? Is the dryness Index a common value which is need to be stated?

**RESPONSE**: Dryness index adopted here is a climate index that was widely used to reflect the degree of dryness in the atmosphere; it often defined as the ratio of potential evaporation to precipitation (Xiao et al., 2013). A dryness index of 15.9 means a very dry climate, under which potential evaporation rate could be ~15 times higher than the precipitation received. The dryness index of 15.9 is a common value for the arid northwestern China, but much higher than the rest regions of China. We have clarified this point in our coming revision. Please see Line 107-108 (Page 3) for details.

9) Lines 111: What exactly is meant with sandy soil? What official soil name does it correspond to? leave out "...coarse texture and....". Scotch or Scots pine? Use familiar expression.

**RESPONSE**: According to Yang and Liu (2010), the zonal soils in our study region are loamy sand and sandy soil, which are two soil types typical for arid and semiarid environments (Zhao et al., 2010), we will further clarify this point in our revision. As suggested, "coarse texture" has been removed from the revision, and "Scotch Pine" was replaced with "Scots pine" too.

10) Lines 124-126: Please explain why the different treatments used in the study were chosen.

**RESPONSE**: As we mentioned at the beginning of this paragraph, the different treatments used in the study were chosen to investigate the accumulative effect of different cropping systems and agronomic manipulation on soil property evolution. More details have been included in the revision to solve this concern.

11) Lines 127: Why using exactly this type of irrigation (furrow irrigation)? Please explain in more detail.

**RESPONSE**: Because it was the most widely used irrigation type in our study area, and even the entire northwestern China. More details have been added in the revision to solve this concern

12) Lines 128: Why just using one site with no film-mulching and five with mulching? Not a sufficient comparison possible between the sites.

**RESPONSE**: Please refer to our response to Question 3.

13) Lines 242: Is it not unrealistic to use a ground level of the soil matric potential, even though the water level never reaches that high up?

**RESPONSE**: I guess this question may be raised by some misleading description about the lower boundary, for example, "i.e., $h = -5cm$". In fact, we adopted a free drainage boundary, which also can be described as "a unit vertical hydraulic gradient boundary condition which can account for a variable flux". To clarify this point, we have reorganized this sentence as "A free-drainage boundary condition was applied along the bottom boundary".

14) Lines 245: Which software? Please specify.

**RESPONSE**: We do this calculation by coding in MATLAB environment. It has been clarified in the revision. Please see page 7 line 249 for details.

15) Lines 343: In what dimension is the soil water content measured?

**RESPONSE**: According to the Operator's Manual, each TDR sensor (5TE, Decagon Devices Inc. Pullman, WA, USA) uses an electromagnetic field (dimensions: 9.3 x 2.4 x 6.5 cm) to measure the dielectric permittivity $\varepsilon$ and thus the soil water content of the surrounding medium. In this study, the TDR sensors were installed at 5 different depths (20, 40, 60, 80, and 100 cm) at each plot, to monitor the soil water moisture of the root zone (0-110 cm).

16) Lines 55-59: The sentence is too long, therefore hard to follow and should be divided into two or three sentences. End the first sentence with a full stop after (Wright, 1971).

**RESPONSE**: We have divided this sentence into two ones as suggested. Please see page 2 line 56-59 in the revision.

17) Lines 92-95: Unorganized sentence order makes it even harder to follow the content.

**RESPONSE**: To make it clearer and more understandable, we have reorganized this sentence as "Exploring a reliable farmland *SWBC* estimation model, which can make the most of the vast amounts of soil moisture data, is crucial for irrigation management optimization in arid regions with coarse-textured soils (Musters and Bouten, 2000;Sharma et al., 2017)". Please see Page 2 Line 93-94 in the coming revision.

18) Lines 165-171: Subdivision into subsets probably better for this sentence.

**RESPONSE**: Changed as suggested.

19) Lines 219-226: Hard to follow the derivation.

**RESPONSE**: We have reorganized this part to make the derivation easier to follow in the revision.

20) Lines 42: missing 'the' ...the Heihe river basin (HRB) is one of the largest...

**RESPONSE**: Corrected as suggested.

21) Lines 81: Cross out "...quite common and..."

**RESPONSE**: Corrected as suggested.

22) Lines 83: Cross out "...and more..."

**RESPONSE**: Corrected as suggested.

23) Lines 85: Also, with this process_, ...

**RESPONSE**: Corrected as suggested.

24) Lines 86: ..., almost no work_ have been...

**RESPONSE**: Corrected as suggested.

25) Lines 112: ... (planted since the 1970s), include Haloxylon anmmodendron, ... →either no comma or 'including'

**RESPONSE**: Corrected as suggested.

26) Figure 1: The figure is not entirely clear→to what do the roots belong on the right?/figure on the left→layout and position of the legend and unprecise placement of the small map of China.

**RESPONSE**: We have reorganized this figure to make it clearer and more understandable in the revision according to this comment.

27) Figure 2: Probably better 'day of year' as axis label instead of DOY.

**RESPONSE**: We have added the explanation in the note. Please see page 6 line 183 in the revision.

28) Lines 140: 'was' instead of 'were'.

**RESPONSE**: Corrected as suggested.

29) Lines 195: 'dominates' instead of 'dominants'.

**RESPONSE**: Corrected as suggested

30) Table 1: we propose to insert this nomenclature-table at the beginning of the chapter.

**RESPONSE**: We decide to keep them at the original places just for the tidy layout.

31) Table2: Vertical lines between wilting point value of one study site and the saturated water conductivity of the next study site could probably increase readability (see attached pdf).

**RESPONSE**: Thanks for the nice suggestion, we have added vertical lines for Table 2 in the revision.

32) Figure 5: Does it need this figure at this place? and what exactly, apart from the clearly visible irrigation events, should be shown with it?

**RESPONSE**: This figure shows all the soil moisture dynamics that we used in this paper to further do our calculations, analysis and discussions.

33) Figure 6: left graph: use other scaling, since nothing is readable/ right graph: why is the scale in the middle of the graph (a bit weird position), why using such fancy boxplots if normal rectangular ones could be used?

**RESPONSE**:  Both the two panels in Figure 6 have been reorganized as suggested in the revision.

34) Lines 437: explains

**RESPONSE**:  Corrected as suggested.

35) Lines 381: 'Literature' instead of 'literatures'?

**RESPONSE**:  Corrected as suggested.

36) Lines 443: ... (Fares and Alva, 2000), suggesting that there is... In general: Reflect on the placement and use of figures and tables in the work, so that these stylistic tools fulfil their purpose of increasing the attractiveness of a scientific paper.

**RESPONSE**:  Added as suggested.

37) Lines 1-2: as the use of soil moisture measurements is a major part of the scientific work, we would adjust the title as follows: Estimation of Evapotranspiration and Other Soil Water Budget Components, Using Soil Moisture Measurements, in an Irrigated Agricultural Field of a Desert Oasis Or also the following possibility seems easier to understand to us: Estimation of Evapotranspiration, Irrigation and Drainage, Using Soil Moisture Measurements, in an Irrigated Agricultural Field of a Desert Oasis

**RESPONSE**:  According the suggestion, we have changed the title as "Estimation of Evapotranspiration and Other Soil Water Budget Components, Using Soil Moisture Measurements, in an Irrigated Agricultural Field of a Desert Oasis".

38) Lines 29-30: Leave out the obvious parts and concentrate on the findings

**RESPONSE**:  Because this part is one of the most important findings and we would prefer to keep in the abstract.

39) Lines 106: lowest and highest temperatures for winter and summer, respectively→that's logical, no need of repetition. Pleonasm. It would be probably better to use the terms 'minimum' and 'maximum' in this context.

**RESPONSE**:  Corrected as suggested.

40) Lines 323-324: water content values are difficult to read in the presented form of a listing.

**RESPONSE**:  We have reorganized this sentence as "For the same interval of time, the water contents in the 40-, 60-, 80- and 100-cm depths of soil decreased from 25.4%, 19.8%, 18.5% and 14.2%, to 15.7%, 14.3%, 15.4% and 12.8%, respectively". Please see Page 10 Line 322-323 in the revision.

41) Lines 512: Setting upper boundaries would have been a nice addition.

**RESPONSE**: Yes, we agree, but we don't have more detailed information to set such a special upper boundary for inter-cropping treatments in this study. However, uncertainty that may be caused by this simplicity have been discussed in our manuscript.

42) Lines 566: It would be desirable for the conclusion to mention what would be appropriate irrigation methods for this variety of agricultural soil.

**RESPONSE**: Good idea, but this is beyond the scope of this article, and we are preparing another paper to discuss this issue.

**References:**

Dudley, L. M., Ben-Gal, A., and Shani, U.: Influence of plant, soil, and water on the leaching fraction, J Environ Qual, 39, 713-724, 2008.

Fares, A., and Alva, A. K.: Evaluation of capacitance probes for optimal irrigation of citrus through soil moisture monitoring in an entisol profile, Irrigation Sci, 19, 57-64, 10.1007/s002710050001, 2000.

Musters, P. A. D., and Bouten, W.: A method for identifying optimum strategies of measuring soil water contents for calibrating a root water uptake model, J Hydrol, 227, 273-286, 2000.

Sharma, H., Shukla, M. K., Bosland, P. W., and Steiner, R.: Soil moisture sensor calibration, actual evapotranspiration, and crop coefficients for drip irrigated greenhouse chile peppers, Agr Water Manage, 179, 81-91, 2017.

Xiao, Z., Yang, F., Shi, F., Nakatsuka, T., and Shi, J.: Comparison of the dryness/wetness index in China with the Monsoon Asia;Drought Atlas, Theoretical & Applied Climatology, 114, 553-566, 2013.

Yang, R., and Liu, W.: Nitrate contamination of groundwater in an agroecosystem in Zhangye Oasis, Northwest China, Environmental Earth Sciences, 61, 123-129, 10.1007/s12665-009-0327-7, 2010.

Zhao, W., Ji, X., Kang, E., Zhang, Z., and Jin, B.: Evaluation of Penman-Monteith model applied to a maize field in the arid area of Northwest China, Hydrology and Earth System Sciences Discussions, 7, 461-491, 2010.

---

## Editor Decision (ED1)

1.      Water budget method is actually not water budget, it is actually a water balance method. Budget term often means storage, but in your method only soil moisture is a budget term, other terms (evaporation, percolation, irrigation) are all flux terms. So it is suitable to name it as water balance method.

2.      The paragraph starting from L62: the author should pay attention to the following work, which has utilized measurement of soil moisture and ET (by Eddy Covariance) to estimate deep percolation. Zhang, Z., Hu, H., Tian, F., Yao, X., and Sivapalan, M.: Groundwater dynamics under water-saving irrigation and implications for sustainable water management in an oasis: Tarim River basin of western China, Hydrol. Earth Syst. Sci., 18, 3951-3967, doi:10.5194/hess-18-3951-2014, 2014.

3.      The paragraph starting from L121, it is redundant to describe the field experiment in terms of cropping effect on soil property, because the main target of this study is to explore the flux estimation by using water budget methods.

4.      The paragraph from L139, more details on irrigation should be given because of its importance, e.g., irrigation method, timing, irrigation quota, during, etc. Table is preferred.

5.      Section 2.3 1), it is unclear how to decide percolation. It is confusing to say deep percolation begins after soil moisture storage reaches its maximum (I assume it is Smax) but to say for a specific case percolation occurs after Smax occurs. It is unclear in Fig2(b) how to determine the timing when percolation starts.

6.      When the authors adopt Richards Equation to calculate slow drainage term, a lot of uncertainty related with soil hydraulic parameters are introduced. How do the authors deal with this uncertainty?

7.      The title should be something like: quantification of soil water balance components based on continuous soil moisture measurement and Richards equation

8.      In 4.3, I don't think it is necessary to discuss the impact of cropping system on soil property. It is irrelevant to the topic.

9.      Line 485 - 487, the statement is true. For the extreme case, irrigation lasts throughout the growing season with very small irrigation intensity, which should mean water can only wet surface thin soil layer and it cannot support water to crop root. Also, you should consider the salinity issue. See the discussion in Gao Long, Tian Fuqiang, Ni Guangheng, Hu Heping. Experimental study on soil water and salt movement and irrigation scheduling for cotton under mulched drip irrigation condition. Journal of Hydraulic Engineering, 2010, 41(12):1158-1165. (in Chinese)

10.     P16: it is hard to understanding the statement 'irrigation is challenge to measure'.

11.     P25, published SWBCs values at nearly sites: how accuracy are the published SWBCs? How are these values obtained?

12.     P54, in this region. The authors should not be specific to Heihe River basin. If you argue the method to be applicable to arid areas, you should discuss the issue in wider areas especially other similar regions like Tarim River Basin in China and Aral Sea Basin in Central Asia as examples. See the following for reference. Fuqiang Tian, You Lu, Hongchang Hu, Wolfgang Kinzelbach & Murugesu Sivapalan (2019): Dynamics and driving mechanisms of asymmetric human water consumption during alternating wet and dry periods, Hydrological Sciences Journal, DOI: 10.1080/02626667.2019.1588972

---

## Author Response (AR3)

**Authors' Responses**

The authors would like to gratefully acknowledge the insightful comments and the encouraging support of the editor, Prof Fuqiang Tian. We also appreciate the time the anonyms reviewers put in reading our manuscript, and the comments were valuable, refreshing, and encouraging. We have taken the time to think through all the comments and have adequately addressed all the comments item-by-item in the following 14 pages. As a result of these suggestions, we believe that the resulting paper is effectively improved. Before we submit the revision, a native English speaker (Marian Rhys, she is also a professional English editor) was invited to polish the manuscript thoroughly. In the following document, we are providing the responses to reviewers, as we have submitted in the interactive discussion. Please find below your reproduced comments, followed by our point-by-point responses (in blue or red). Below the responses to reviewers, we are providing the marked-up revised document.

**Response to Editor**

**General comments**

**As you can see from Referees' comments as well as my own comments, a significant revision is still required. Please revise your manuscript by strictly following all comments.**

RESPONSE: We warmly thank Professor Fuqiang Tian for the overall favorable impression of the work, and for his thorough review and the detailed, helpful comments. Please find our point-by-point responses.

**Especially, I would like to draw the authors' attention to the points below:**
**1) Be careful with the term used, especially water budget, drainage driver, etc.**

RESPONSE: Thanks for the nice suggestion. We agree that misusing the terms could lead to potential confusions for readers. According to the suggestion, "water budget" has been replaced with "water balance" and the "drainage driver" also has been removed from the revision.

**2) The major work is the application of inverse Richards approach to obtain drainage and ET. Please clarify its novelty.**

RESPONSE: To solve the concern, the introduction section was partly reworded and additional information about the novelty of this work were added (Line 81-104). We also clarify and highlighted the novelty in the discussion section (4.4 Potential for SWBC estimation by using soil moisture measurements). Please check the revised manuscript for details (Line 491-496, Line 519-521).

**3) Highlight other possible novelties.**

RESPONSE: We pointed out in the introduction section of the revision that "These types of measurements provide critical information for ecohydrology, agricultural, and hydrological researches in the arid environments, but mostly served as either an indicator for drought monitoring and forecasting (Anderson et al., 2012), or boundary conditions and/or calibration data for models (Vereecken et al., 2008). So far, however, relatively few work has been published on testing the potential of using a soil moisture database as a method to systematically estimate the *SWBCs* of farmland in the drylands, where the principal soils are coarse-textured (Grayson et al., 1999;Yang et al., 2018b) and tend to have low water retention capacity and higher drainage (Lal, 2004)". We also highlighted that "…exploring a reliable farmland *SWBC* estimation method that can make the most of the vast amounts of soil moisture data, is crucial for irrigation management optimization (Musters and Bouten, 2000; Sharma et al., 2017), especially for irrigating arid regions with such coarse-textured soils." Please check the revised manuscript for details. (Line 491-496, Line 519-521).

**Specific comments**

**1. Water budget method is actually not water budget; it is actually a water balance method. Budget term often means storage, but in your method only soil moisture is a budget term, other terms (evaporation, percolation, irrigation) are all flux terms. So, it is suitable to name it as water balance method.**

RESPONSE: Thanks for the nice suggestion, we fully agree with this point. Following the suggestion, we changed the term of "water budget method" to "water balance method" in this revision.

**2. The paragraph starting from L62: the author should pay attention to the following work, which has utilized measurement of soil moisture and ET (by Eddy Covariance) to estimate deep percolation. *Zhang, Z., Hu, H., Tian, F., Yao, X., and Sivapalan, M.: Groundwater dynamics under water-saving irrigation and implications for sustainable water management in an oasis: Tarim River basin of western China, Hydrol. Earth Syst. Sci., 18, 3951-3967, doi:10.5194/hess-18-3951-2014, 2014.***

RESPONSE: We have carefully read the recommended literature, and got many inspires from it. The paper published by Zhang et al. (2014) also was cited in this revision as an case study that calculate deep percolation as a residual of the water balance (See Line 65-67 in the revision for details).

**3. The paragraph starting from L121, it is redundant to describe the field experiment in terms of cropping effect on soil property, because the main target of this study is to explore the flux estimation by using water budget methods.**

RESPONSE: This paragraph has been slightly shortened to avoid redundancy, and make it more focused on the main target which is to explore the flux estimation by using water budget methods (See Line 128-132 in the revision).

**4. The paragraph from L139, more details on irrigation should be given because of its importance, e.g., irrigation method, timing, irrigation quota, during, etc. Table is preferred.**

RESPONSE: Although most of the details have been included in the early version of the manuscript. More detailed information (i.e., irrigation method, irrigation quota, etc.) were added as a new table in the revision according to this suggestion (Please see Line 150-155 in the revision for details). A new table was also added to show the irrigation schedule and quota (Table 1). please see Line 168-171 for details.

**5. Section 2.3 1), it is unclear how to decide percolation. It is confusing to say deep percolation begins after soil moisture storage reaches its maximum (I assume it is $S_{max}$) but to say for a specific case percolation occurs before $S_{max}$ occurs. It is unclear in Fig.2(b) how to determine the timing when percolation starts.**

**RESPONSE**: Sorry for the confusing wording. The detailed methods used to determine percolation were descripted in Section 2.3 2), and in Section 2.3 1), we provided an assumption here that "no surface-water excess or steady-state flow took place during any irrigation event, and deep percolation usually occurred after soil moisture storage reached maximum ($S_{max}$) …". This assumption is the base of percolation determination in next step. We further clarified this point in the revision and the confusing sentence has been reorganized based on the suggestion provided by Professor Tian as below: "Although a few specific cases of percolation could occur before the $S_{max}$ is reached (second panel in Fig. 2b), it had little effect on the estimation of irrigation volume because the maximum soil water storage differed little (by only 1.86 mm) before and after deep percolation began. For instance, we checked all the irrigation events of NT1-NT6 during the entire growing season period, and there were no underestimates of $S_{max}$ except for two irrigation events in NT2, which only had a slight underestimate of 1.86 mm and 10.3 mm, and generated errors of 1.1% and 4.1%, respectively". The timing when percolation starts in Fig.2(b) was determined whenever the soil water content in the deepest layer (90-110 cm) is measured to be greater than "field capacity" ($\Theta_{fc}$), i.e., Rice et al. (1986)". We have added this information in the revision, please see Line187-194 for details.

**6. When the authors adopt Richards Equation to calculate slow drainage term, a lot of uncertainty related with soil hydraulic parameters are introduced. How do the authors deal with this uncertainty?**

**RESPONSE**: This is a very important question. Yes, potential uncertainties could be introduced into the calculation of slow drainage term by soil hydraulic parameters, due to the possible changes over time in the parameters itself (Shein, 2015). This point has been considered when preparing the earlier version of the manuscript, so that all the parameters, including soil bulk density ($\rho_b$), vertical saturated hydraulic conductivity ($K_s$), and soil water retention, were determined using standard laboratory procedures on undisturbed soil cores in steel cylinders taken at 20-cm intervals down to 100-cm depth. The measured parameters were further profile-averaged for all the plots to improve the experimental uncertainty. To better solve the concern, we have clarified this point and some related discussions were added in the revision (see 4.5 Uncertainty analysis for details). We also argued "Although this cannot fully prevent the development of uncertainty caused by the parameters, such uncertainties are trivial especially in light of the relatively small proportion of slow drainage in the context of sandy soils, i.e., only about 9.5% of the drainage occurred during this stage according to our calculation (Table 3)". Please see Line 539-543 in the revision for details.

**7. The title should be something like: quantification of soil water balance components based on continuous soil moisture measurement and Richards equation.**

**RESPONSE**: According to the comments, the title of this paper has been improved to "*Quantification of Soil Water Balance Components Based on Continuous Soil Moisture Measurement and Richards Equation in an Irrigated Agricultural Field of a Desert Oasis*". Thanks for the nice suggestion.

**8. In 4.3, I don't think it is necessary to discuss the impact of cropping system on soil property. It is irrelevant to the topic.**

**RESPONSE**: We agree that this part is a little bit irrelevant to the main topic present in this paper. After careful consideration and discussion with the coauthors, we reorganized this section as a new one "4.3 Effects of the variances in soil hydrophysical properties on the SWBC estimation". Most of the irrelevant discussions and conclusions have been removed from the revision, and potential problems that could be introduced into the SWBC estimation by the variances in soil hydrophysical properties were kept and elaborated in the revision. We stressed that "In this desert oasis and other ones located in arid northwest China, most of the fields belong to smallholder farmers, who usually follow different cropping patterns and tillage methods, resulting in a heterogeneity of soil hydrophysical properties. For the soil-moisture data-based method proposed in this paper, the spatial heterogeneity of the soil hydrophysical properties may restrict its applicability to a large agricultural area". Through a brief analysis of the potential influence of different cropping systems on soil hydrophysical properties, we confirmed that "different cropping systems and agronomic manipulation have limited effects on differing soil physical characteristics in sandy soil, at least at a decade scale", and we argued that "The limited influence of different cropping systems on soil hydrophysical properties in coarse-textured soil environments at a 10-year scale indicates a good stability and representativeness of the measured soil hydrophysical data and thus a good application prospect for applying the soil-moisture data-based method in practice." Please see Line 461-480 in the revision for details.

**9. Line 485 - 487, the statement is true. For the extreme case, irrigation lasts throughout the growing season with very small irrigation intensity, which should mean water can only wet surface thin soil layer and it cannot support water to crop root. Also, you should consider the salinity issue. See the discussion in Gao Long, Tian Fuqiang, Ni Guangheng, Hu Heping. Experimental study on soil water and salt movement and irrigation scheduling for cotton under mulched drip irrigation condition. Journal of Hydraulic Engineering, 2010, 41(12):1158-1165. (in Chinese)**

RESPONSE: Very nice suggestion. To be more logical and rational, this statement has been rewording as: "Early researches suggested that decreasing the irrigation amount and increasing the irrigation frequency, and thus maintaining a relatively constant level of soil moisture with less stress from "too little or too much", is the potential choice for saving water and improving water use efficiency in arid regions like the middle HRB". We have read the recommended paper and some related ones, and mentioned the salinity issue in this part of the revision, i.e., "This method could also contribute to alleviate salt accumulation in agricultural soils and sustain ability of irrigated lands in arid regions by providing key *SWBCs* information for farmers and other decision makers in agricultural production (Gao et al., 2010)." Please see Line 497-500 in the revision for details.

**10. P16: it is hard to understanding the statement 'irrigation is challenge to measure'.**

RESPONSE: Sorry for the confusing wording. What I want to express is that "*Site-specific irrigation is challenging to measure*", the reason is that: "the two most common methods of measuring irrigation water—water meters or indirect methods—pose both economic and operational challenges to water managers, due to the wide spatial distribution of small fields throughout rural areas (Folhes et al., 2009)". To solve the concern, we have reorganized the related statement in our revision. Please see Line 61-64 in the revision for details.

**11. P25, published SWBCs values at nearby sites: how accuracy are the published SWBCs? How are these values obtained?**

RESPONSE: Table 4 has been reorganized to be clearer and more logical. The calculating methods or models of the published ET values at nearby sites have been given in the Table 4 in the earlier version of manuscript, and more details of the published SWBCs were further included in this table of the revised manuscript. Unfortunately, we cannot find the accuracy of the reported values due to lacking more detailed data in the references.

**12. P54, in this region. The authors should not be specific to Heihe River basin. If you argue the method to be applicable to arid areas, you should discuss the issue in wider areas especially other similar regions like Tarim River Basin in China and Aral Sea Basin in Central Asia as examples. See the following for reference.** *Fuqiang Tian, You Lu, Hongchang Hu, Wolfgang Kinzelbach & Murugesu Sivapalan (2019): Dynamics and driving mechanisms of asymmetric human water consumption during alternating wet and dry periods, Hydrological Sciences Journal, DOI: 10.1080/02626667.2019.1588972*

RESPONSE: Thanks for the constructive suggestion and useful literature. We elaborated this part, and added some more discussions about the methods' applicability in other similar regions. Please see Line 491-496 in the revision for details. The nice reference has been cited to ascertain its importance [i.e., Tian et al. (2019)].

**Response to Anonymous Referee #1 (RC1)**

**General comments**

The manuscript uses timeseries of soil moisture data to estimate the soil water budget components, especially evapotranspiration, in an irrigated agricultural filed of a desert oasis. This study is well conducted, and the authors responded the comments well. I think it is worth publishing after minor revisions. I have several comments listed as below.

RESPONSE: We warmly thank the Anonymous Referee #1 for the overall favorable impression of the work, and for his or her thorough review and the detailed, helpful comments. Please find below your reproduced comments, followed by our point-by-point responses.

**Specific comments**

1) The anonymous reviewer 3# concerns the locations of Section 3.1, 3.2, and 3.3, and the authors insisted the former locations. I think the problem may be arisen by the title of them. The current titles may be misleading to about the dataset, nor the observed or calculated results. For example, Section 3.2 is about calculated irrigation amount. The meteorological data should be introduced in the materials section.

RESPONSE: We have reorganized the sub-titles of Section 3 as: "3.1 Soil hydrophysical characteristics", "3.2 Soil moisture dynamics (SMDs)", and "3.3 Soil water balance components (SWBCs)". The results about irrigation amount has been merged into section "3.3 Soil water balance components (SWBCs)", and the descriptions upon meteorological data has been moved to "2.2 Site description". Thanks to the nice suggestion, this part looks much better than before.

2) I am confused by the $S_{stop}$ and $S_{max}$. Is $S_{stop}$ larger than $S_{max}$, as shown in Fig. 2.

RESPONSE: Very nice question, and the answer is: $S_{max}$ is larger than $S_{stop}$. As we mentioned in "2.3 Calculation methods 1) Water storage and irrigation amounts", $S_{max}$ was defined as the recorded maximum soil water storage of the root zone, and $S_{stop}$ is the recorded soil water storage when irrigation event ends (moisture of uppermost soil layer starts to decrease). Although the real water storage in the entire root-zone soil should keep constant during the short periods between irrigation ends and deep drainage starts, it is not naturally been recorded by the soil moisture sensors at any time of this period, because of the continuously redistributing soil water profile and limited number of soil moisture sensor. In this work, the real water storage in root-zone soil was assumed to be equal to $S_{max}$, and thus $S_{stop}$ would tend to approach $S_{max}$ if more soil moisture sensors were installed in the soil profile. To clarify this point, more detailed explanations on this point has been included in the revision. Please see the caption of Figure 2 for details in the revision.

3) Please check the captions appeared in the text. For example, Fig.7 appears earlier than Fig. 5. Where is Fig. 6?

**RESPONSE:** We feel very sorry for the careless. "Fig. 7" appeared in Section 3.2 has been replaced by "Table 3" in the text, and we further checked all the figure captions to avoid any other similar mistakes.

4) Fig.3, It is better to show the specific irrigation amount and compare it with rainfall event.

**RESPONSE**: Thanks for the suggestion, Fig.3 has been reorganized to show the average amount of each irrigation event during 2016 in the revision. We also corrected an error of the irrigation date in this figure. We feel very sorry for the careless data checking and glad that we find the mistakes and corrected it in this revision.

**Response to Anonymous Referee #2 (RC2)**

We thank the anonymous Referee #2 for taking the time to review our manuscript and for their generally positive feedback on our study. Please find below your reproduced comments, followed by our point-by-point responses.

**Specific comments**

1) Lines 14, 61 etc.: drainage is not, and has been never considered as a "driver" of hydrological cycle. Irrigation is taken as a factor that interferes hydrological cycle, and seldomly taken as a "driver". You may call evapotranspiration a driver of the hydrological cycle. It is a component of the hydrological cycle as a matter of fact. Usually, the drivers of the hydrological cycle refer to the climatic factors.

**RESPONSE**: Thanks for the nice suggestion, we have changed "driven" as "dominated", and changed "driver" as "components".

2) Lines 314, 315. "Because the inverse method proposed by Zuo et al. (2002) and Guderle and Hildebrandt (2015) had never been applied throughout an entire growing season for farmland...", this is hard to say.

**RESPONSE**: We have removed this sentence in the revision to solve the concern.

3) As indicated in the last paragraph of the introduction section, this work aims to investigate performance of the inverse method to the coarse-textured soils. Thus, "coarse-textured soils" should be focused and highlighted in the discussion. This might have been something new in this paper.

**RESPONSE:** Nice suggestion. Some related discussion will be added in the Section 4.2 and 4.5 in the coming revision to highlight the "coarse-textured soils". Please see "4.2 Accuracy of the other estimated SWBCs" in the revision for details.

**Response to Dr. Michael W. I. Schmidt (SC1)**

**General comments**

*A note upfront from the submitting person:

This review was prepared by Basil Frefel and Michèle Bösiger, both master students in geography at the University of Zurich. The review was part of an exercise during a second semester master level seminar on "the biogeochemistry of plant-soil systems in a changing world", which I organize. We would like to highlight that the depth of scientific knowledge and technical understanding of these reviewers represents that of master students. We enjoyed discussing the manuscript in the seminar, and hope that our comments will be helpful for the authors.*.

Addressing sustainable irrigation in semi-desert regions, Li et al. observed in a soil moisture time series the soil-water holding ability in the Heihe River Basin, northern China. Soil properties such as saturated hydraulic conductivity and soil retention capabilities were combined with the soil water balance method and the inverse Richards equation (water movement in unsaturated soil) in order to estimate the Soil Water Base Components (evapotranspiration, irrigation, drainage). The measurements were taken at six sites in sandy soils, which differed in agricultural technique (rotational, permanent cultivation), plant species and mulching application. The results show that the estimation of the Soil Water Base Components corresponds to the soil-moisture time series and thus should be helpful for future irrigation planning.

Overall, the issue addressed by Li et al. is of great importance regarding future water management. Especially with increasing water scarcity, the study lays the foundation for a sustainable water strategy for one of the biggest agricultural producers worldwide for rice and wheat and thus is a valid contribution to the present scientific knowledge. The article is well structured, and the thinking steps are described profoundly, in addition to the reflected approach to the results. However, we have also found some caveats.

**RESPONSE**: We would like to thank Dr. Michael W. I. Schmidt for organizing the seminar to discuss our manuscript. We also warmly thank Basil Frefel, Michèle Bösiger for their thoughtful review, and for the specific suggestions, with which our manuscript is significantly improved in both its clarity and organization. We have taken the time to think through all the review comments and have adequately addressed all the comments item-by-item in the following pages.

In general, the type size is too small, and the sentences tend to be too long and interlaced to comprehend the article at the first read (see further comments).

**RESPONSE**: 1) Following the reviewer's suggestion, we have carefully revised our manuscript and these long sentences that may be unclear or confusing to a reader were rewritten in the revision.

Moreover, the exact time span of the experiment is unclear, and it should be mentioned what (important) role sandy soils play in the agricultural production of China and the world.

**RESPONSE**: The long-term experiment that was design to investigate the accumulative effect of different cropping systems and agronomic manipulation on soil property evolution was set up in 2007, and will run as long as the funding allows, and the in-situ soil moisture measurements were carried out since 2015, and was designed to continue until the long-term field experiment is ended. We have clarified this point in our revision. In addition, more information about "what (important) role sandy soils play in the agricultural production of China and the world" was included in this revision.

Besides, the abstract should be shortened to stay attractive for the reader and abbreviations such as NT1-NT6 should not be used in this very first paragraph. In our opinion, many explanations are too complicated; Keep explanations short and simple. In contrast to the detailed abstract, the conclusion is too short, again overloaded with abbreviations and therefore not understandable standing alone. More detailed remarks are listed below.

**RESPONSE**: As suggested, we slightly shortened the Abstract and removed the abbreviations such as NT1-NT6 in this part, and try our best to make the explanations as concise as possible. Also, the Conclusion part has been slightly expanded to solve the reviewer's concern. All the abbreviations that could be ambiguous also were carefully defined in the revision. More details can be found in our below responses to specific comments.

**Specific comments**

1) Lines 12: SWBC: In this paper the term soil water budget components and its abbreviation SWBC is used as if it were a standard term in soil science or hydrology. is this really the case? Otherwise it should be mentioned that this term as such is only used in this paper.

**RESPONSE**: Thanks for the nice suggestion. No, this abbreviation is not a standard term both in soil science and hydrology, and it is only used in this paper. We have defined that in the Introduction and then to use it in the following text to reduce repeated use of the term and thus to make the paper as concise as possible. According to the suggestion, we have mentioned that "the abbreviation is used here for simplicity, and effective only in this paper" when it first appeared in the revision.

2) Lines 22: Since the inverse Richards equation is of great importance in this work, but is not necessarily known to the general public, a brief description of what this method is used for, e.g. in parentheses, would be helpful.

**RESPONSE**: Thanks for pointing out this issue. A brief description of what this method is used for have been added in our revision, i.e., "(which is a model of unsaturated soil water flow based on the Richards equation)".

3) Lines 26: Why only one site without film-mulch? Comparison to other sites without mulching would have been helpful.

**RESPONSE**: As we mentioned in Section 2.2, this long-term field experiment was set up in 2007, and the six experiment plots have very different treatments, i.e., from NT1 to NT6, they were sequentially set as: (1) continuous pasture cropping, (2) continuous maize cropping, (3) continuous maize cropping with straw return, (4) maize-maizepasture rotation, (5) maize-pasture rotation, (6) maize-pasture intercropping. In general, only the maize fields need to be film-mulched, and maize were planted in all the plots except NT1 (which was planted with alfalfa in the growing season of 2016), so that NT1 is the only one plot without film-mulch in this study.

4) Lines 29: What should be special about an obvious correlation between the volume of irrigation and drained water? is this really a significant result of the study or could this statement also be omitted?

**RESPONSE:** Yes, we do think this is a significant result of the study. Although similar results were also reported by other works, we found that this linear positive correlation is particularly noticeable in the coarse textured soils like the desert oasis in arid China, as we also discussed in Section 4.2. We think this is a useful result for further improve irrigation strategies, and thus would like to keep it here.

5) Lines 47: What is a high leaching fraction? Explain.

**RESPONSE**: Leaching fraction (LF) represent the ratio of the actual depth of drainage to the depth of irrigation (Dudley et al., 2008), we have explained it in the coming revision. Please see Line 48-49 in the revision for details.

6) Lines 40-54 and 55-69: The second paragraph is redundant.

**RESPONSE**: The second paragraph has been slightly shortened to eliminate any redundant in the revision.

7) Lines 106-107: Reference is needed for the sentence "The annual average precipitation..."

**RESPONSE**: Added as suggested.

8) Lines 107: What does a dryness index of 15.9 mean. Please put this number into context. Is this a high or a low value compared to the surrounding region or the rest of China? Is the dryness Index a common value which is need to be stated?

**RESPONSE**: Dryness index adopted here is a climate index that was widely used to reflect the degree of dryness in the atmosphere; it often defined as the ratio of potential evaporation to precipitation (Xiao et al., 2013). A dryness index of 15.9 means a very dry climate, under which potential evaporation rate could be ~15 times higher than the precipitation received. The dryness index of 15.9 is a common value for the arid northwestern China, but much higher than the rest regions of China. We have clarified this point in our revision.

9) Lines 111: What exactly is meant with sandy soil? What official soil name does it correspond to? leave out "...coarse texture and....". Scotch or Scots pine? Use familiar expression.

**RESPONSE**: According to Yang and Liu (2010), the zonal soils in our study region are loamy sand and sandy soil, which are two soil types typical for arid and semiarid environments (Zhao et al., 2010), we will further clarify this point in our revision. As suggested, "coarse texture" has been removed from the revision, and "Scotch Pine" was replaced with "Scots pine" too.

10) Lines 124-126: Please explain why the different treatments used in the study were chosen.

**RESPONSE**: As we mentioned at the beginning of this paragraph, the different treatments used in the study were chosen to investigate the accumulative effect of different cropping systems and agronomic manipulation on soil property evolution. More details have been included in the revision to solve this concern.

11) Lines 127: Why using exactly this type of irrigation (furrow irrigation)? Please explain in more detail.

**RESPONSE**: Because it was the most widely used irrigation type in our study area, and even the entire northwestern China. More details have been added in the revision to solve this concern.

12) Lines 128: Why just using one site with no film-mulching and five with mulching? Not a sufficient comparison possible between the sites.

**RESPONSE**: Please refer to our response to Question 3 of SC1.

13) Lines 242: Is it not unrealistic to use a ground level of the soil matric potential, even though the water level never reaches that high up?

**RESPONSE**: I guess this question may be raised by some misleading description about the lower boundary, for example, "i.e., $h = -5$cm". In fact, we adopted a free drainage boundary, which also can be described as "a unit vertical hydraulic gradient boundary condition which can account for a variable flux". To clarify this point, we have reorganized this sentence as "A free-drainage boundary condition was applied along the bottom boundary".

14) Lines 245: Which software? Please specify.

**RESPONSE**: We do this calculation by coding in MATLAB environment. It has been clarified in the revision.

15) Lines 343: In what dimension is the soil water content measured?

**RESPONSE**: According to the Operator's Manual, each TDR sensor (5TE, Decagon Devices Inc. Pullman, WA, USA) uses an electromagnetic field (dimensions: 9.3 x 2.4 x 6.5 cm) to measure the dielectric permittivity ε and thus the soil water content of the surrounding medium. In this study, the TDR sensors were installed at 5 different depths (20, 40, 60, 80, and 100 cm) at each plot, to monitor the soil water moisture of the root zone (0-110 cm).

16) Lines 55-59: The sentence is too long, therefore hard to follow and should be divided into two or three sentences. End the first sentence with a full stop after (Wright, 1971).

**RESPONSE**: This sentence has been deleted from in the revision (including the related reference), due to the potential redundancy with the previous paragraph.

17) Lines 92-95: Unorganized sentence order makes it even harder to follow the content.

**RESPONSE**: To make it clearer and more understandable, we have reorganized this sentence as "Exploring a reliable farmland *SWBC* estimation model, which can make the most of the vast amounts of soil moisture data, is crucial for irrigation management optimization in arid regions with coarse-textured soils (Musters and Bouten, 2000;Sharma et al., 2017)".

18) Lines 165-171: Subdivision into subsets probably better for this sentence.

**RESPONSE**: Changed as suggested.

19) Lines 219-226: Hard to follow the derivation.

**RESPONSE**: We have reorganized this part to make the derivation easier to follow in the revision.

20) Lines 42: missing 'the' ...the Heihe river basin (HRB) is one of the largest...

**RESPONSE**: Corrected as suggested.

21) Lines 81: Cross out "...quite common and..."

**RESPONSE**: Corrected as suggested.

22) Lines 83: Cross out "...and more..."

**RESPONSE**: Corrected as suggested.

23) Lines 85: Also, with this process_, ...

**RESPONSE**: Corrected as suggested.

24) Lines 86: ..., almost no work_ have been...

**RESPONSE**: Corrected as suggested.

25) Lines 112: ... (planted since the 1970s), include Haloxylon anmmodendron, ... →either no comma or 'including'

**RESPONSE**: Corrected as suggested.

26) Figure 1: The figure is not entirely clear: to what do the roots belong on the right? /figure on the left→layout and position of the legend and unprecise placement of the small map of China.

**RESPONSE**: We have reorganized this figure to make it clearer and more understandable in the revision according to this comment.

27) Figure 2: Probably better 'day of year' as axis label instead of DOY.

**RESPONSE**: We have added the explanation in the note. Please see page 6 line 183 in the revision.

28) Lines 140: 'was' instead of 'were'.

**RESPONSE**: Corrected as suggested.

29) Lines 195: 'dominates' instead of 'dominants'.

**RESPONSE**: Corrected as suggested

30) Table 1: we propose to insert this nomenclature-table at the beginning of the chapter.

**RESPONSE**: We decide to keep them at the original places just for the tidy layout.

31) Table2: Vertical lines between wilting point value of one study site and the saturated water conductivity of the next study site could probably increase readability (see attached pdf).

**RESPONSE**: Thanks for the nice suggestion, we have added vertical lines for Table 2 in the revision.

32) Figure 5: Does it need this figure at this place? and what exactly, apart from the clearly visible irrigation events, should be shown with it?

**RESPONSE**: This figure shows all the soil moisture dynamics that we used in this paper to further do our calculations, analysis and discussions, so that we would like to keep it in the manuscript.

33) Figure 6: left graph: use other scaling, since nothing is readable/ right graph: why is the scale in the middle of the graph (a bit weird position), why using such fancy boxplots if normal rectangular ones could be used?

**RESPONSE**: Both the two panels in Figure 6 have been reorganized to solve the concerns, however, the original box and Whisker plot (with diamond patterns) was kept in this revision, because this design would make the figure looks much better.

34) Lines 437: explains

**RESPONSE**: Corrected as suggested.

35) Lines 381: 'Literature' instead of 'literatures'?

**RESPONSE**: Corrected as suggested.

36) Lines 443: ... (Fares and Alva, 2000), suggesting that there is... In general: Reflect on the placement and use of figures and tables in the work, so that these stylistic tools fulfil their purpose of increasing the attractiveness of a scientific paper.

**RESPONSE**: Added as suggested.

37) Lines 1-2: as the use of soil moisture measurements is a major part of the scientific work, we would adjust the title as follows: Estimation of Evapotranspiration and Other Soil Water Budget Components, Using Soil Moisture Measurements, in an Irrigated Agricultural Field of a Desert Oasis Or also the following possibility seems easier to understand to us: Estimation of Evapotranspiration, Irrigation and Drainage, Using Soil Moisture Measurements, in an Irrigated Agricultural Field of a Desert Oasis

**RESPONSE**: According the suggestion, we have changed the title as "Estimation of Evapotranspiration and Other Soil Water Budget Components, Using Soil Moisture Measurements, in an Irrigated Agricultural Field of a Desert Oasis".

38) Lines 29-30: Leave out the obvious parts and concentrate on the findings

**RESPONSE**: Because this part is one of the most important findings and we would prefer to keep in the abstract.

39) Lines 106: lowest and highest temperatures for winter and summer, respectively→that's logical, no need of repetition. Pleonasm. It would be probably better to use the terms 'minimum' and 'maximum' in this context.

**RESPONSE**: Corrected as suggested.

40) Lines 323-324: water content values are difficult to read in the presented form of a listing.

**RESPONSE**: We have reorganized this sentence as "For the same interval of time, the water contents in the 40-, 60-, 80- and 100-cm depths of soil decreased from 25.4%, 19.8%, 18.5% and 14.2%, to 15.7%, 14.3%, 15.4% and 12.8%, respectively".

41) Lines 512: Setting upper boundaries would have been a nice addition.

**RESPONSE**: Yes, we agree, but we don't have more detailed information to set such a special upper boundary for inter-cropping treatments in this study. However, uncertainty that may be caused by this simplicity have been discussed in our manuscript.

42) Lines 566: It would be desirable for the conclusion to mention what would be appropriate irrigation methods for this variety of agricultural soil.

**RESPONSE**: Good idea, but this is beyond the scope of this article, and we are preparing another paper to discuss this issue.

[revised manuscript text omitted]